# Behavioral and brain- transcriptomic synchronization between the two opponents of a fighting pair of the fish *Betta splendens*

Trieu-Duc Vu [1,2,3,4,5◉], Yuki Iwasaki [2,6◉], Shuji Shigenobu [7], Akiko Maruko [1,2], Kenshiro Oshima [1,2], Erica Iioka [2], Chao-Li Huang [8], Takashi Abe [9], Satoshi Tamaki [10], Yi-Wen Lin [5], Chih-Kuan Chen [4], Mei-Yeh Lu [4], Masaru Hojo [5], Hao-Ven Wang [5], Shun-Fen Tzeng [5], Hao-Jen Huang [5], Akio Kanai [10], Takashi Gojobori [11], Tzen-Yuh Chiang [5], H. Sunny Sun [12], Wen-Hsiung Li [4,13], Norihiro Okada [1,2,5,6]*

1 School of Pharmacy, Kitasato University, Tokyo, Japan, 2 Foundation for Advancement of International Science, Tsukuba, Japan, 3 Life Sciences and Biotechnology Dept, Tokyo Institute of Technology, Tokyo, Japan, 4 Biodiversity Research Center, Academia Sinica, Taipei, Taiwan, 5 Department of Life Sciences, National Cheng Kung University, Tainan, Taiwan, 6 Nagahama Institute of Bio-Science and Technology, Nagahama, Japan, 7 National Institute for Basic Biology, Okazaki, Japan, 8 Institute of Tropical Plant Sciences, National Cheng Kung University, Tainan, Taiwan, 9 Graduate School of Science and Technology, Niigata University, Niigata, Japan, 10 Institute for Advanced Biosciences, Keio University, Yamagata, Japan, 11 Computational Bioscience Research Center, King Abdullah University of Science and Technology, Thuwal, Kingdom of Saudi Arabia, 12 Institute of Molecular Medicine, National Cheng Kung University, Tainan, Taiwan, 13 Department of Ecology and Evolution, University of Chicago, IL, United States of America

◉ These authors contributed equally to this work.
* okadano@pharm.kitasato-u.ac.jp

**Data Availability Statement:** The RNA-Seq data are accessible on DDBJ (https://www.ddbj.nig.ac.jp/index-e.html) with this ID: DRA009599.

## Abstract

Conspecific male animals fight for resources such as food and mating opportunities but typically stop fighting after assessing their relative fighting abilities to avoid serious injuries. Physiologically, how the fighting behavior is controlled remains unknown. Using the fighting fish *Betta splendens*, we studied behavioral and brain-transcriptomic changes during the fight between the two opponents. At the behavioral level, surface-breathing, and biting/striking occurred only during intervals between mouth-locking. Eventually, the behaviors of the two opponents became synchronized, with each pair showing a unique behavioral pattern. At the physiological level, we examined the expression patterns of 23,306 brain transcripts using RNA-sequencing data from brains of fighting pairs after a 20-min (D20) and a 60-min (D60) fight. The two opponents in each D60 fighting pair showed a strong gene expression correlation, whereas those in D20 fighting pairs showed a weak correlation. Moreover, each fighting pair in the D60 group showed pair-specific gene expression patterns in a grade of membership analysis (GoM) and were grouped as a pair in the heatmap clustering. The observed pair-specific individualization in brain-transcriptomic synchronization (PIBS) suggested that this synchronization provides a physiological basis for the behavioral synchronization. An analysis using the synchronized genes in fighting pairs of the D60 group found genes enriched for ion transport, synaptic function, and learning and memory. Brain-transcriptomic synchronization could be a general phenomenon and may provide a new cornerstone with which to investigate coordinating and sustaining social interactions between two interacting partners of vertebrates.

**Funding:** This study was supported by National Cheng Kung University (NCKU), Taiwan with support from the Aim for the Top University Project of NCKU (D104-38A05 & D105-38A03) and the Ministry of Science and Technology (NSC 102-2621-B-006-002- and MOST 103-2621-B-006-005-) to N.O. and from the Ministry of Science and Technology, Taiwan (MOST 107-2311-B-001-016-MY3) to W-H.L. It was also supported by the Japan Society for the Promotion of Science (JSPS) Grants-in-Aid for Scientific Research, KibanB to N.O. The funders had no role in study design, data collection and analysis, decision to publish, or preparation of the manuscript.

**Competing interests:** The authors have declared that no competing interests exist.

## Author summary

Agonistic encounters induce changes in the brain and behavior, but their underlying molecular mechanisms remain poorly understood. The fighting fish *Betta splendens* are small freshwater fish that are well known for their aggressiveness and are widely used to study aggression. Here, by measuring aggressive behavior displays (bite/strike/surface-breathing) between two opponents during fighting, we demonstrate that the two opponents in each fighting pair showed similar fighting configurations by influencing each other. In addition, we compared brain gene expression between opponents and showed synchronization of gene expression within a fighting pair, leading to pair-specific synchronization in genes associated with ion transport, synapse function, and learning and memory. This study presents the possibility that similar behaviors in pairs of animals under similar conditions may trigger synchronizing waves of transcription between the individuals, providing a hint to support the idea that fighting behaviors contain cooperative aspects at the molecular level.

## Introduction

Animal contests result in an unequal division of resources and thus are a principal driver of natural selection [1]. Conspecifics animals fight for resources such as food and mating opportunities, but they usually stop fighting after assessing their relative fighting abilities to avoid serious injuries [2–4]. The contest outcomes have been commonly explained by the mutual assessment model [5], which, for example, was demonstrated in the case of zebrafish [6] and in the fish *Betta splendens* [7]. Yet, how such mutual assessments are carried out at the molecular level during fighting remains unknown.

Aggression is ubiquitous in animals because it may increase an individual's chance to survive and to transmit its genes to the next generation [8]. Various types of aggression in animals have been described [9]. The fighting fish *B. splendens* is well known for its long fighting duration. It belongs to the suborder Anabantoidei of the perciform ray-finned freshwater fish, which are distinguished by their possession of a lung-like labyrinth organ that helps them to carry out surface-breathing [10–12]. This fish has been commonly used to study the biological mechanism of aggression for various reasons. *B. splendens* males are extremely aggressive and have stereotypical social displays [13, 14] (S1A Fig). In their natural habitats, males fiercely protect their territories, where they build a bubble nest to hold fertilized eggs [15]. In the laboratory, a male aggressively attacks any intruder or its own mirror image to maintain its territory [14, 16]. Previous studies on *B. splendens* and other fish focused on aggressive displays between males but usually described only one particular behavior (e.g., displaying, biting, or striking) [17, 18] or compared global differences in aggressiveness between the loser and the winner [19–23].

More recently, physiological and neurological studies have been conducted to examine aggressive behaviors using *B. splendens* in combination with pharmacological treatments, and some important signaling molecules linked to aggression have been discovered, such as serotonin (5-HT) [24], dopamine [25], and GABA [26]. Additionally, a suite of genes linked to aggressiveness has been identified in different fish such as cichlid fish [27], zebrafish [28], and rainbow trout [29]. However, these analyses focused on differentially expressed genes (DEGs) in the brains of the eventual winner and loser, revealing only the differences in the brain-transcriptomic state between them without comparing the brain-transcriptomic responses of the

two opponents in a fighting pair during a fight. As a result, no comprehensive molecular analysis of the brain-transcriptomic state of male-male competition during a fight in *B. splendens* has been attempted at the genomic level. Two major obstacles to such studies were the large number of genes known to influence aggression in animals [30, 31] and the lack of a reference genome for this fish, which was released only recently [32].

Behavioral and physiological synchronization have long been studied by scientists given its importance in reproductive/mating behavior, cooperative behavior, aggressive behavior, etc. [33, 34]. Crucial signaling molecules such as 5-HT and arginine-vasotocin (AVT) modulate the levels of such cooperation at the physiological level in fish [35–37]. Furthermore, recent studies have provided physiological evidence for such synchronization at the neuronal level between two interacting bats or mice [38, 39]. Given that clear behavioral evidence over the course of several decades as well as the recent physiological findings described above, we hypothesized that synchronization at the gene expression level between two interacting partners, especially in brain genes associated with synaptic function, ion transport, and aggression, might also occur. We specifically ask the following questions 1) what is the relationship between brain-transcriptomic activity in two individuals that are engaged in social interactions? 2) how does this relationship change across different timescales to facilitate social interactions?

We evaluated this possibility by examining the behavioral and brain-transcriptomic changes between two interacting opponents of *B. splendens* during fighting. Males of this fish undergo a stress response during a 15-min fight [14, 16]. Over a similar time period, differences in the expression of early genes as well as other socially responsive genes have been detected in various fish [40–47]. Accordingly, we examined a 20-min and a 60-min fighting period to correlate the brain transcriptome responses with behavioral changes [44–47]. Whereas the 20-min fight allowed us to capture immediate early genes (IEGs)-known as genomic markers for brain activity and markedly new experiences for fish, the 60-min fight permitted us to evaluate brain-transcriptomic responses for the full range of volitional behavior in response to an opponent over the course of the encounter. We found the presence of a unique temporal fighting structure by comparing the frequencies of each behavior (i.e., biting/striking, mouth-locking, and surface-breathing) of the two opponents in the intervals between mouth-lockings over the time course of a fight. Also, we discovered the synchronization of the brain transcriptomes between the opponents in a fighting pair by testing for correlation values of gene expression between paired fish and unpaired fish. Furthermore, we identified and examined the synchronized gene sets by analyzing DEGs, gene ontology and gene pathway enrichment to understand what kinds of brain activity might underlie the transcriptomic synchronization during competitive interaction.

## Results

### Behavioral synchronization achieved through fighting interaction

To examine the behavioral changes of the two opponents of a fighting pair, we selected male fish and allowed them to fight in pairs for a duration of 20 min (D20) or 60 min (D60) or until the conflicts were resolved (typical fighting pairs) (Fig 1A and 1B). The data obtained from 17 typical fighting pairs revealed that the fighting process in *B. splendens* proceeded in a stereotypic manner: surface-breathing (started at 0.94 ± 0.24 min), biting/striking (3.49 ± 0.60 min) and mouth-locking (10.60 ± 1.12 min) (S1 Table, S1 Video). During a 60-min fight, the mouth-locking events occurred an average of 12.12 ± 2.40 times, and each event lasted for 1.29 ± 0.24 min; therefore, mouth-locking events accounted for ~26.88% of the 60 min. Although the fighting duration varied dramatically among the pairs, surface-breathing, biting/

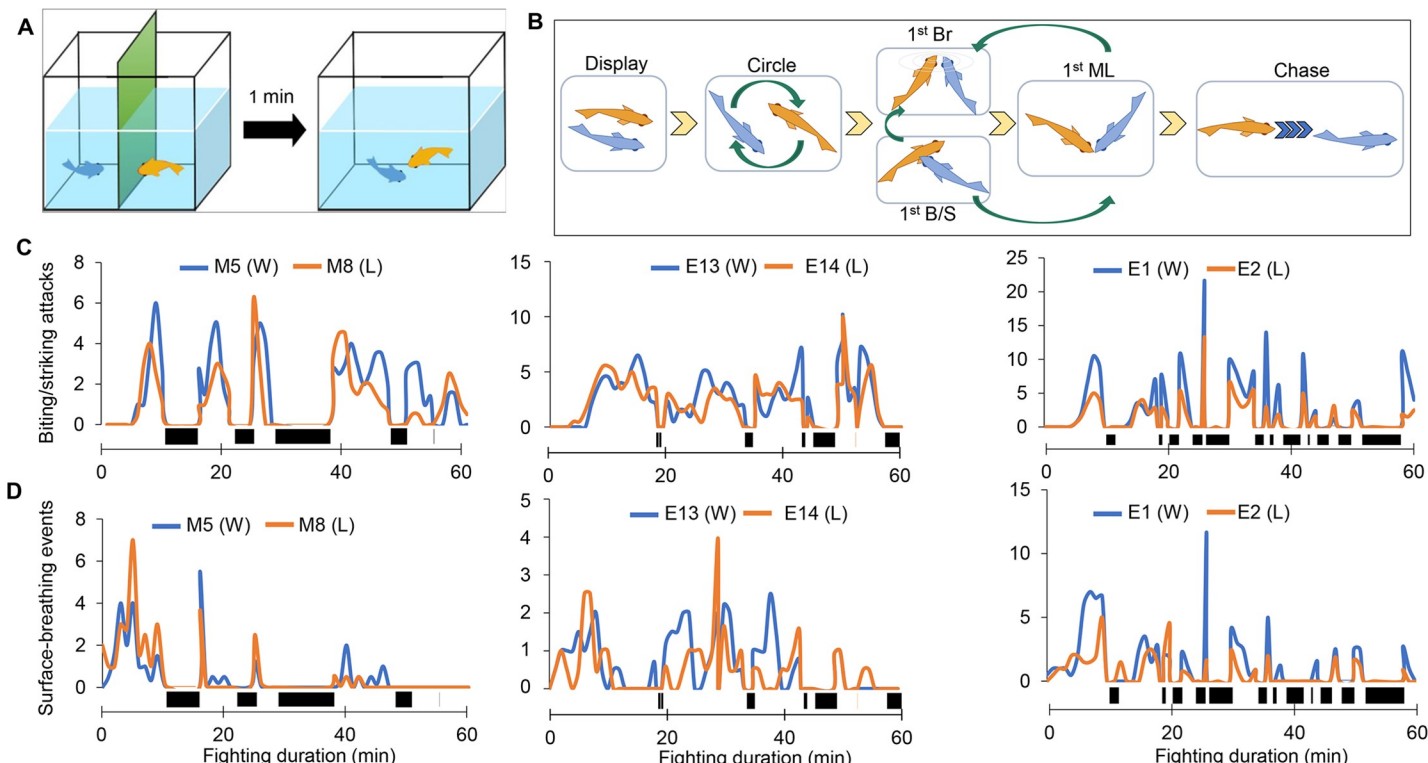

**Fig 1. Schematic representation and behavioral synchronization. (A)** Setup of the fighting experiment. **(B)** Schematic illustration of dynamic fighting behaviors between two male fish. The fighting duration and the first occurrence of each specific behavior are recorded. Abbreviations: Br, surface-breathing event; B–S, biting and striking attack; and ML, mouth-locking. **(C)** Behavioral differences among all fighting pairs with respect to biting/striking attack frequency. **(D)** Behavioral differences among all fighting pairs with respect to surface-breathing frequency. Behavioral analysis of two opponents in a pair; the data were obtained from three pairs, M5 vs. M8, E13 vs. E14, and E1 vs. E2 with respect to biting/striking attacks and surface-breathing events. Blue lines indicate the frequencies of behaviors of the ultimate winner, and dark-yellow lines indicate those of the ultimate loser in a fighting pair. Black bars indicate mouth-locking events and durations, during which biting/striking and surface-breathing behaviors are restricted, n = 6.

striking, and mouth-locking occurred in this order among all fighting pairs, showing a temporal fighting structure.

To understand in detail the fighting configuration, a 60-min fighting period for eight of the 17 typical fighting pairs was analyzed in 2-min windows with a 1-min overlap in terms of the frequency and duration of surface-breathing and biting/striking (see Methods). Among pairs, the patterns of surface-breathing and biting/striking were very similar between the opponents of each fighting pair (Fig 1C and 1D, S2A–S2J Fig), because these behaviors occur only between two consecutive mouth-lockings. Accordingly, the frequency, duration, and distribution of mouth-lockings during the fighting period determined the fight structure, and each fighting pair showed a pair-specific fighting pattern. More specifically, the frequencies of biting/striking and surface-breathing behaviors of the two opponents showed a high correlation for each of the eight pairs (S2 Table and S3 Table).

## Brain-transcriptomic response triggered by the fighting experience

To examine the transcriptome changes during a fight, we conducted RNA-seq analysis of 25 whole-brain RNA samples, including those from five non-fighting individuals (i.e., before fighting group, B), from five pairs after a 20-min fight (the D20 group), and from five pairs after a 60-min fight (the D60 group) (Fig 2A; S4 Table). A principle component analysis (PCA) and a linear discriminant function analysis (LDA) from the 25 libraries were done on

the same dataset—top 50% most variable gene transcripts (11,653 genes) to differentiate the fighting groups (see Methods). While the PCA clearly illustrated that paired fish in the D60 group were closely clustered together (S4E Fig), the LDA nicely separated the fighting groups with two significant functions: Function 1 (LD1), which separated the D60 group from the D20 and non-fighting group, explained 69.1% of the variance, and Function 2 (LD2), which separated the non-fighting group from the D60 and D20 groups, explained 30.9% of the variance (Fig 2B). A high consistency among the brain transcriptome profiles induced by each social experience (B, D20 and D60) indicates that a particular brain-transcriptomic state (co-regulated gene sets) underlies each behavioral state.

To evaluate the brain transcriptomic differences between the D20 and D60 groups, we tested for DEGs. Comparing each social treatment (i.e., D20 or D60) with the non-fighting group (B) revealed a total of 1,082 DEGs (FDR < 0.05, |log FC| > 2), with a greater number of up-regulated genes than down-regulated genes in both fighting groups (Fig 2C). Noticeably, FDR means false discovery rate and FC refers to fold change. Whereas 859 and 518 genes showed significant increases, only 40 and 46 genes showed significant decreases in the D60 and D20 groups, respectively (Fig 2D). A plot in which the D20 DEGs are projected onto the D60 DEGs and vice versa revealed that most of the genes with down-regulated expression in the D20 group were also down-regulated in the D60 group and that those with up-regulated expression in the D20 group remained up-regulated in the D60 group. There were only three genes (unannotated) that were down-regulated in the D20 but were up-regulated in the D60, whereas just one gene (unannotated) was up-regulated in the D20 but was down-regulated in the D60 (Fig 2E).

Among 1,082 DEGs, 564 genes were specifically expressed in the D60 group (D60-specific genes), 223 genes were specifically expressed in the D20 group (D20-specific genes) and 295 genes were expressed in both the D20 and the D60 groups (common genes) (Fig 3A, S5 Table). We found that the probability of 295 overlapping DEGs between the D20 and D60 groups was greater than that expected by chance (p < 7.2e-298, hypergeometric test). In addition to the attributes of their expression across fighting stages, several of these DEGs genes have previously known to associate with neuronal activity (e.g., *c-fos*, *fosb*, *egr4*), neural plasticity processes related to learning and memory (e.g., *bdnf*, *npas4*, *nr4a1*), neurotransmitter (e.g., *grin2a*, *grik2*) and epigenetic function (*rnf44*) (Fig 3B and see Discussion) [6, 48]. The heatmap generated from all 1,082 DEGs illustrated three main patterns of gene expression that were distinguishable among the fighting groups (B, D20, and D60) (Fig 3C). We also note that a large set of unannotated DEGs was uncovered in this analysis (S5 Table).

## Genes associated with each fighting group (D20 and D60)

A <u>w</u>eighted <u>g</u>ene <u>c</u>o-expression <u>n</u>etwork <u>a</u>nalysis (WGCNA) was used to find gene clusters (modules) among all 23,306 genes across all 25 samples (see Methods). Our goal was to identify modules associated with the fighting groups (D20 and D60). This analysis identified 37 co-expression gene modules, which are represented by different colors as named by the program (Fig 4A, S4D Fig). To relate these gene modules to the fighting groups (D20 or D60), associations between the eigengene of each gene module and the fighting groups were computed. The results showed that the two fighting groups were associated with different modules. Particularly, the D20 samples were characterized by an up-regulation of the MEmidnightblue gene module, which included 186 genes that are associated with transcription regulation (Fig 4B, S3A Fig and S3F Fig). The D60 group was characterized by three modules: an up-regulation of the MEgrey60 module (168 genes), which is enriched in genes related to stress response, transcription regulation, and MAPK signaling (Fig 4B, S3B Fig and S3G Fig); an up-regulation of

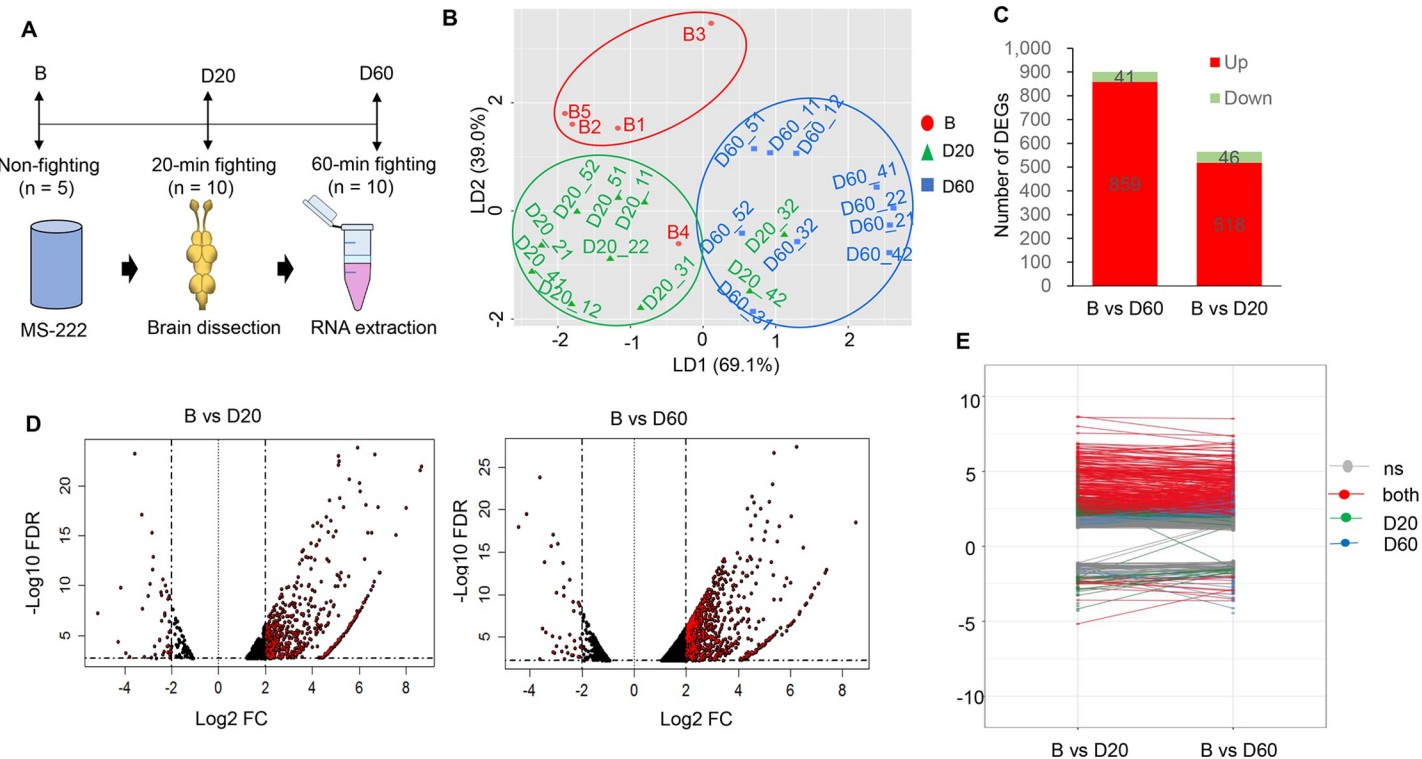

**Fig 2. Brain-transcriptomic response. (A)** Overview of experimental design. **(B)** Clustering of all 25 brain samples using the linear discriminant function analysis (LDA); blue, non-fighting group (B); red, D20 group; and green, D60 group. **(C)** Bar graph showing the number of DEGs generated from the comparisons of B vs. D60 and B vs. D20. Significantly upregulated and downregulated DEGs are represented in red and green, respectively. **(D)** Volcano plot of the DEGs obtained from the B vs. D20 and B vs. D60 comparisons. Vertical lines indicate the threshold for a relative expression fold change (FC) of 2 or −2 as compared with controls (B). The horizontal line represents the threshold of a 0.05 FDR value. The red points were significantly upregulated or downregulated in B vs. D20 and B vs. D60. **(E)** A plot in which the D20 DEGs are projected onto the D60 DEGs and vice versa; 'ns' means not significant, 'both' means significant in both D20 and D60 groups. Blue, significant only in the D20; green, significant only in the D60.

the MEturquoise module (6,390 genes), which is enriched in genes related to phosphorylation, ion transport, protein transport, and MAPK signaling (Fig 4B, S3D Fig and S3I Fig); and an down-regulation of the MEblue module (4,256 genes), which is enriched in genes related to translation, metabolic process, DNA-repair, and MAPK signaling (Fig 4B, S3C Fig and S3H Fig).

We found that the D20-specific genes, D60-specific genes, and common genes in the DEG analysis appeared in different modules in this WGCNA analysis. The D20-specific genes were present in the MEbrown (104 genes), MEred (86 genes), MEgreen (5 genes), MEdarkgreen (4 genes), and other gene modules (S6 Table). The D60-specifc genes were present in the MEturquoise (444 genes), MEblue (29 genes), MEgrey60 (18 genes), MElightgreen (17 genes), and other gene modules (S6 Table). The common genes were present in the MEbrown (170 genes), MEturquoise (79 genes), MEred (20 genes), and other modules (S6 Table). The DEG and WGCNA analyses used different approaches but showed a high agreement in the characterization of the transcriptome changes associated with each fighting group across the fight.

## Brain-transcriptomic synchronization occurs in a pair-specific manner

We discovered the presence of brain-transcriptomic synchronization by first testing for the correlation coefficients (r values) of expressed gene transcripts between the paired fish and unpaired fish over the 23,306 gene transcripts for the D20 (D20-paired fish vs. D20-unpaired

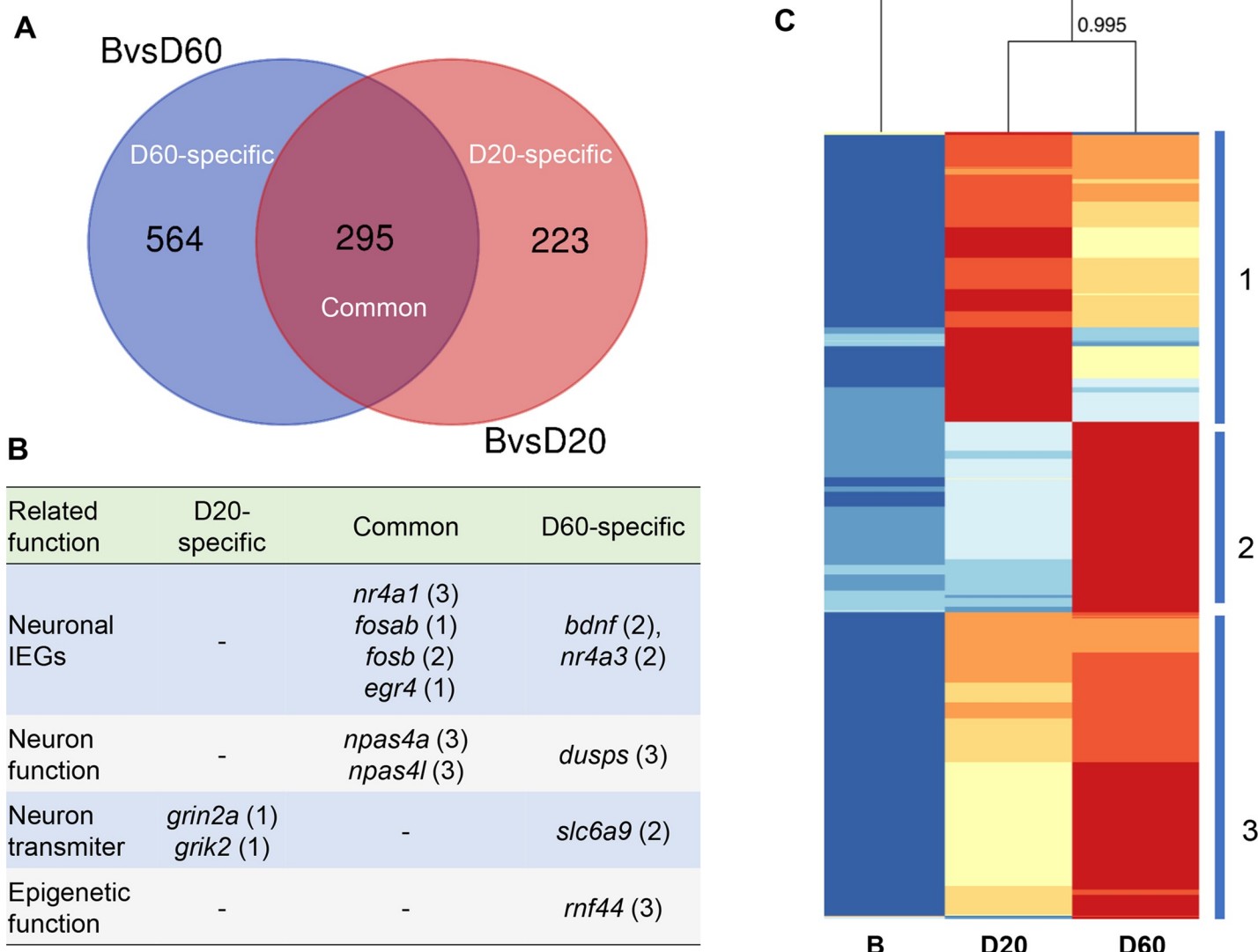

**Fig 3. Characterization of DEGs. (A)** Venn diagram displaying the number of common and specific DEGs identified in the B (non-fighting group) vs. D60 and B vs. D20 comparisons. **(B)** Representative genes among the DEGs that are related to neuronal functions; the numbers in the parentheses indicate which clusters in the heatmap these genes belong to; hyphen '-' means not found. **(C)** Heatmap using all 1,082 DEGs illustrates four main patterns of gene expression within each of the three groups (B, D20, and D60). Similarities between the fighting groups as shown by hierarchical clustering can be seen above the heatmap. The bootstrap value at the node was obtained by the *hclust* function in R. Intensity of color indicates expression levels: red, high expression; blue, low expression. The numbers (1−3) on the right side refer to the three main patterns of gene expression.

fish) and D60 (D60-paired fish vs. D60-unpaired fish) fighting groups (S4A Fig and S4B Fig). The results revealed that the r values for paired fish were significantly higher than those for unpaired fish in both groups (permutation test, $p < 0.05$), especially for the D60 group (permutation test, $p = 0.001$) (Fig 5A). Furthermore, the clustering analysis in the heatmap using the top 10% most variable genes (2,330 genes) showed that all five pairs of the D60 group were clustered together, in sharp contrast to the single pair that clustered together in the D20 group (Fig 5B). It is interesting to note that similar results were obtained when individual profiles of all the 25 samples using all gene set (23,306 genes) were clustered (S5C Fig).

Next, we subsequently examined the fine structure of brain-transcriptomic synchronization in each fighting pair by applying the grade of membership (GoM) model to the 25 brain

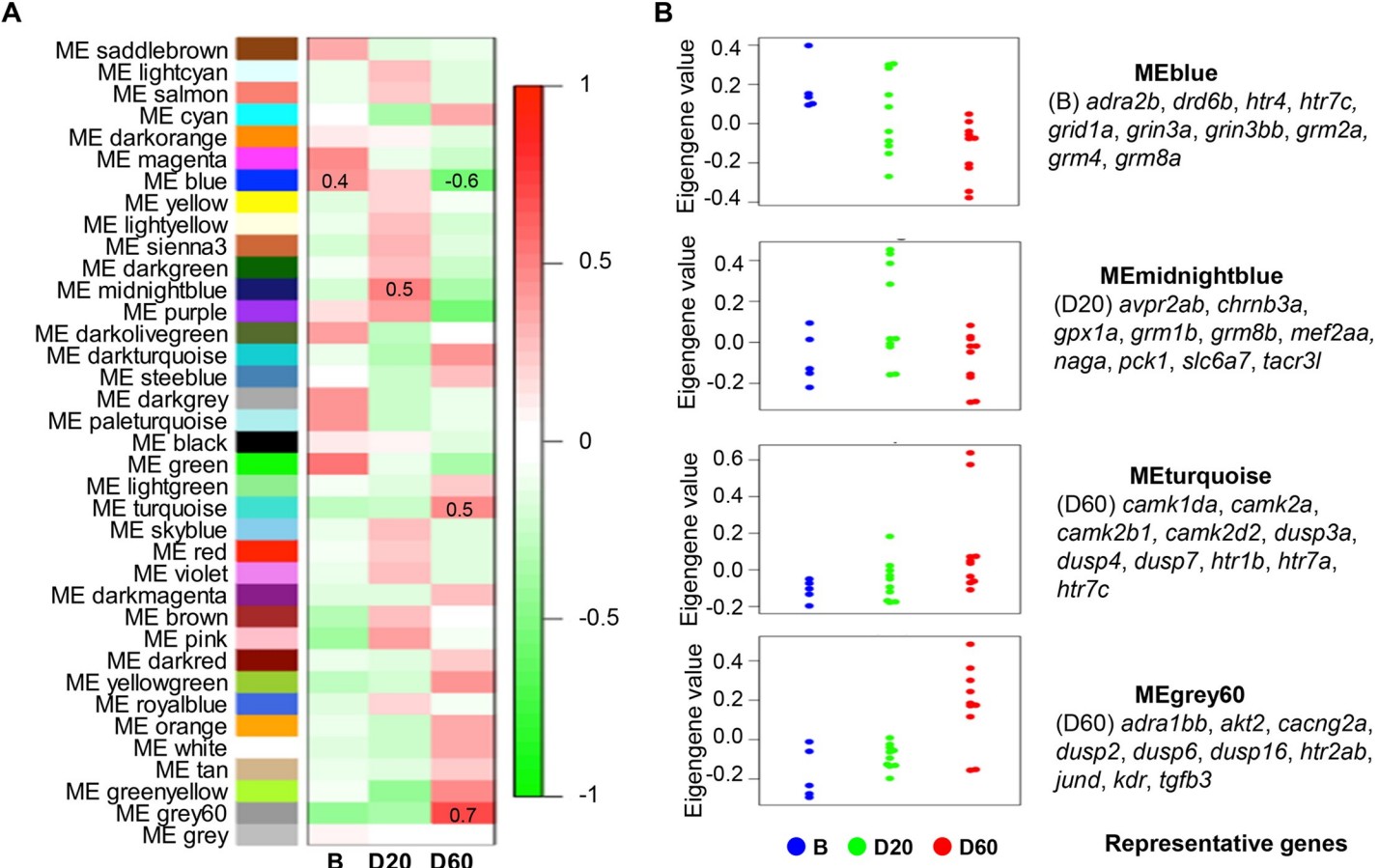

**Fig 4. Co-expression gene modules for different fighting groups (B, D20, and D60). (A)** Associations between patterns of expression in the 37 identified modules across all three groups (B, D20, and D60). The colors of the boxes are scaled with the value of the correlation coefficients, ranging from red (r = −1) to green (r = 1). **(B)** Eigengene values of samples separated by group (B, D20, and D60) for gene modules significantly associated with four groups (blue, midnight blue, turquoise, and grey60). Representative genes enriched in each module are shown.

samples, each of which included the 23,306 gene transcripts. This GoM model allowed us to highlight similarities among the samples by determining their similar membership (cluster) proportions (see Methods). Interestingly, the composition of clusters for individuals in each pair from the D60 group was very similar, and moreover the pattern from any given pair of the D60 group was unique relative to that of the other pairs. This observation suggested that brain-transcriptomic synchronization is achieved according to the specific interactions between the two opponents of a pair, such that a particular set of genes was upregulated or downregulated and synchronized in a pair-specific manner (Fig 5C). The top five driving genes (i.e., the genes that were most distinctively differentially expressed) in each cluster were extracted from the total gene transcripts and were found to be unique for each cluster (S7 Table). Altogether, these findings suggest that the brain-transcriptomic state of the two opponents in a pair were mutually influenced during fighting and became synchronized in a pair-specific manner during the fighting process. This synchronization was weak in the D20 group but became strong in the D60 group.

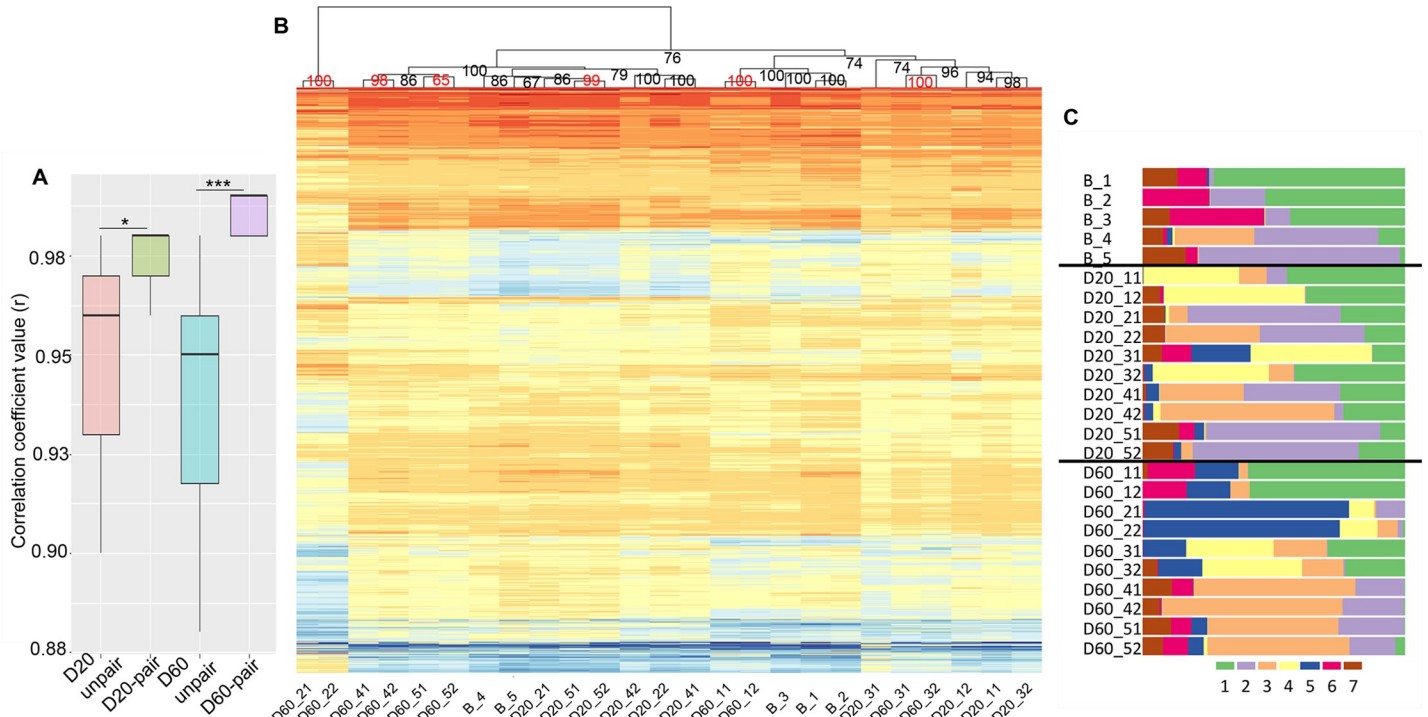

**Fig 5. The structure of brain-transcriptomic synchronization. (A)** Boxplot of correlation coefficient values (r) for comparisons of the TMMs for 23,306 gene transcripts between paired fish and unpaired fish from the same treatments (D20-paired vs. D20-unpaired and D60-paired vs. D60-unpaired, permutation test). $^*$p $< 0.05$; $^{***}$p $< 0.001$. **(B)** Heatmap using the 25 cDNA libraries with each of the 2,330 gene contigs (top 10% most variable genes). Intensity of color indicates expression level (dark orange, high expression; sky blue, low expression). Similarities between individuals within fighting pairs and between fighting pairs as shown by hierarchical clustering can be seen above the heatmap. Bootstrap values at the nodes were obtained by *hclust* function in R. Bootstrap values in red indicate the two opponents in a pair were clustered together. **(C)** GoM analysis using TMM values of all 23,306 genes generated from the 25 libraries with a set of seven clusters represented by different colors.

## Characterization of synchronized genes

To identify synchronized genes, we conducted a two-step analysis (see Methods). First, we obtained 868 and 2,409 DEGs from the comparison of the D20 and D60 groups relative to the non-fighting group, respectively (FDR $\leq 0.05$ and log FC $> 0$) (Fig 6A, S8 Table). Here we focused on the up-regulated gene sets because the number of these genes was much higher than that of the down-regulated genes among the DEGs in both groups (Fig 2C), and we used a less stringent criteria than was used in the analysis of Fig 2C (log FC $> 0$ instead of |log FC| $>2$) so that more DEGs will be included for the downstream analysis. Next, from these DEGs, we extracted 172 and 1,522 genes that were synchronized in at least one pair from the D20 and the D60 group, respectively, according to the criteria defined in the Methods (Fig 6A, S9 Table). Comparisons between the two synchronized gene sets showed that the number of synchronized genes within one pair, two pairs, three pairs, four pairs, and five pairs was much higher in the D60 group than the D20 group (Fig 6B). The composition of these synchronized gene sets confirmed that brain-transcriptomic synchronization is a major process because more than half (63.2%) of the total DEGs were involved in this process. Moreover, almost half of the synchronized genes were synchronized in five fighting pairs (Fig 6C). It is interesting to note, however, that the synchrony levels were different among each particular fighting pair as shown for neuronal activity−associated genes such as *bdnf*, *calm2a*, and several other genes (Fig 6D, S4C Fig). The overlapping synchronized genes among the five pairs in the Venn diagrams show the number of genes involved in brain-transcriptomic synchronization across all

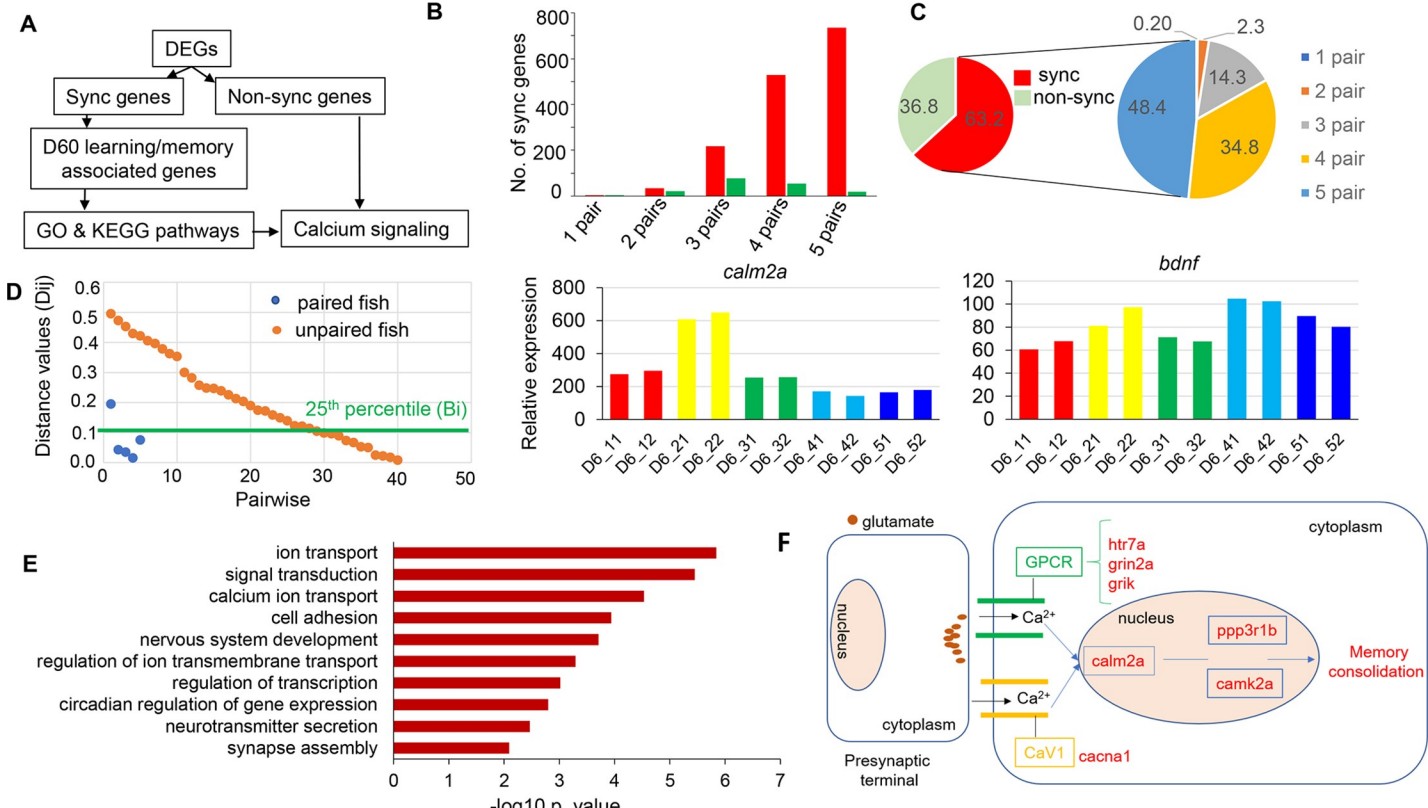

**Fig 6. Synchronization associated with learning and memory genes. (A)** Workflow illustrating the GO and KEGG pathway analyses. (**B**) Bar graph showing the number of synchronized genes in one, two, three, four, and five pairs in the D60 and D20 groups. (**C**) Pie charts showing proportions of synchronized genes. (Left), the percentage of synchronized vs. non-synchronized genes; (right), synchronized genes in one, two, three, four, and five pairs; the data here represent pooled data for both groups (**D**) Examples of genes showing pair-specific individualization of brain-transcriptomic synchronization. (Left), representation for identification a baseline (Bi) of a particular gene i in pair (s) j. The gene i demonstrated here is *camk1da* from the D60 group. Each dot represents a pairwise (the x-axis shows the total of 45 pairwise from the D60 group); blue represents the D60 paired fish; orange represents the D60 unpaired fish; the green line refers to the 25th percentile rank. (Center), *bdnf* represents IEGs and (right), *calm2a* represents memory related genes. (**E**), Significantly enriched biological processes among the synchronized genes in the D60. (**F**) Simplified calcium signaling pathway map generated from the D60 synchronized genes (see S6A Fig and Discussions for detailed information). The up-regulated synchronized genes involved in memory consolidation are shown in red. GPCR, G protein–coupled receptor; CaV1, L-type calcium channel.

five fighting pairs as well as those that were shared between each particular fighting pair in both the D20 (S5A Fig) and D60 (S5B Fig) fighting groups.

To gain insight into the biological functions of synchronized genes involved in brain-transcriptomic synchronization, we first examined the overrepresentation of the Gene Ontology (GO) terms related to biological process (BP) within the 1,282 synchronized genes (all of which were well annotated) in the D60 using DAVID ($p < 0.05$) (Fig 6A). Accordingly, 49 overrepresented BP terms were enriched (S10 Table); however, as the GO terms obtained from DAVID seemed to be redundant, we next used the REVIGO tool to filter these redundant BP terms. Subsequently, 31 overrepresented BP terms remained (S10 Table). Interestingly, this synchronized gene set was mainly enriched in terms related to neurotransmitter secretion, ion transport, and synapse function (Fig 6E, S10 Table). We also looked for overrepresentations among molecular function (MF) and cellular component (CC) terms. Enriched MF terms were associated with transcription factor activity, receptor activity, and ion channel activity (S11 Table), whereas enriched CC terms were associated with membrane, synapse, and AMPA glutamate receptor complex (S12 Table). Furthermore, overrepresentation analysis of KEGG pathways ($p < 0.05$) within 1,282 synchronized genes revealed 19 pathways that were

significantly enriched and are known to have causal effects on social behavior such as calcium signaling [49], MAPK signaling [50], GnRH [51], and others (S13 Table).

## Brain-transcriptomic synchronization associated with long-term memory genes

In mice and bats, social interactions between two interacting partners induce inter-brain correlation of neural activity in the prefrontal cortex of their brains [38, 39], this activity has been implicated in supporting working memory in human [52]. Learning and memory are crucial processes in the winner or loser effect phenomenon, in which fish increase or decrease their likelihood of winning or losing in a future fight after experiencing previous wins or losses, respectively [53]. This motivated us to examine whether genes related to long-term memory were involved in the brain-transcriptomic synchronization in the D60 group.

We first extracted 378 genes from the 1,522 synchronized genes of the D60 group that were linked to learning or memory (see Methods). Next, these 378 genes were used for a KEGG pathway enrichment analysis using DAVID (Fig 6A). As a result, 19 pathways were significantly enriched such as insulin signaling pathway, calcium signaling pathway, adrenergic signaling pathway, etc. (p < 0.05, S13 Table). Among the top significantly enriched pathways shown in S13 Table, the calcium signaling pathway, which was ranked second when pathways were sorted by p-values, is well known for its involvement in learning and long-term memory formation [49]. In support of this finding, several genes in this pathway were up-regulated in the route leading to learning and memory (Fig 6F, S6A Fig).

Next, we used the calcium pathway as a proxy for learning and memory examination for other gene sets including 104 synchronized genes in the D20, 567 non-synchronized genes in the D20, and 791 non-synchronized genes in the D60 to see if they were involved in pathways related to learning and memory (Fig 6A). By comparing the gene expression of those involved in the calcium pathways generated from the KEGG analysis, we found that the D20 synchronized gene set was not enriched for the calcium pathway, and none of the genes involved in calcium pathways generated from the non-synchronized gene sets were included in the pathway of learning and memory (S6B Fig and S6C Fig). Collectively, these results suggest that learning and memory may play a key role in brain transcriptomic synchronization after the fish experience a long period of fighting (see Discussion).

## Discussion

### Behavioral synchronization in vertebrates

In the case of competition among conspecific male animals for resources and mates, they typically stop fighting after assessing their relative fighting abilities to avoid serious injuries. How animals assess another's ability remains poorly understood. Here, we demonstrated behavioral synchronization in each *B. splendens* pair during fighting. Although the behavior sequences (displaying, biting, and striking) in *B. splendens* are in accordance with other fish such as zebrafish (*Danio rerio*) [53] and cichlid fish (*Nannacara anomala*) [54], characteristic pair-specific behavioral synchronization as defined by the pattern of mouth-locking behavior is observed only in *B. splendens*, underscoring the unique behavior of this fish. During fighting, it seems that each *B. splendens* male in a fighting pair attempts to increase its chance of winning by imposing its aggressive behaviors upon its opponent, leading to tightly synchronized behaviors. A few previous studies have found synchronization in other animal behaviors such as foraging (seeking food) and cooperative hunting [55–58]. For example, during foraging by a pair of sticklebacks or Atlantic salmon fish, if there is an advantage to foraging together, the

behavior of both individuals becomes synchronized; however, when one of the two fish receive additional food, they tend to behave independently.

Our behavior analysis showed the presence of a specific behavioral synchronization pattern for each fighting pair. It is likely that each individual of *B. splendens* has its own behavioral characteristics, referred to as "personality" (e.g. shy, bold, proactive, or reactive) [59]. Accordingly, the characteristics of the synchronization pattern for each pair might be influenced by the personality of the individual fish involved (e.g., how shy or bold they are), as each *B. splendens* individual needs to modify its behavior to match the ability of its competitor. This explanation is consistent with previous studies in other fish such as guppies and sticklebacks and in pair-bonded birds such as the great tits (*Parus major*), in which the frequency and distribution of their interactions depend on their body length, color, and proactive or passive exploration personality [60, 61].

## A model for behavioral and brain-transcriptomic synchronization

To explain how behavioral synchronization and brain-transcriptomic synchronization occur, we propose the model presented in Fig 7, in that the mutual assessment could be involved. The repeated attempts of each opponent to match the actions of the other opponent lead to behavioral and then transcriptomic synchronization. At the behavioral level, the two fish initiate and escalate the fight (Fig 7A). Their interactions provide signals for the mutual assessment process to assess the opponent's fighting ability. Moreover, to match the attacks of its opponent, each fish adjusts its fighting behavior to achieve synchronization.

At the physiological level (Fig 7B), the signals of mutual assessment in each fighting individual trigger a transcriptomic response in neurons leading to changes in the physiology of neurocircuits. The first transcriptional wave after neuronal activation induces rapid transcription of immediate-early genes (IEGs) such as *c-fos* [43]; brain-derived neurotrophic factor, *bdnf* [62]; and nuclear receptor transcription factor subfamily *nr4a(1,3)* [63]. Then, a downstream genomic response cascade takes place with the involvement of synchronized genes (S9 Table) and learning- and/or memory-associated genes such as *calm2a*, *camk2b1*, *ppp3r1b*, and others [64], eventually leading to the pair-specific individualization of brain-transcriptomic synchronization (PIBS). These processes are mediated by second messengers, intracellular pathways such as the MAPK pathway [50], ligand-receptor pathways, and/or calcium signaling pathways [64]. Collectively, PIBS presents a mechanism for coordinating and sustaining interactions between opponents.

Although our data are correlative and many details of the proposed model at the molecular level remain unclear, the model may provide a useful basis for future study. It will be interesting to see if both behavioral and brain-transcriptomic synchronization also occur during other interactive behaviors.

## Insights into brain-transcriptomic synchronization

We demonstrated the presence of brain-transcriptomic synchronization, which was observable to a lesser extent after a 20-min fight and was strengthened after a 60-min fight in accordance with behavioral synchronization between fighting pairs of *B. splendens*. We characterized it as follows.

First, the DEG analysis illustrated three main patterns of gene expression that reflected changes in the transcriptomic landscapes between the social groups (B, D20, and D60; Fig 3C), with the involvement of IEGs in both fighting groups. As mentioned above, some IEGs exhibited strong upregulation in the D60 group such as *cfos*, *bdnf*, *nr4a3*, and *neurotransmitter transporter glycine member 9* (*slc6a9*) [65] (Fig 3B), which are involved in long-term memory

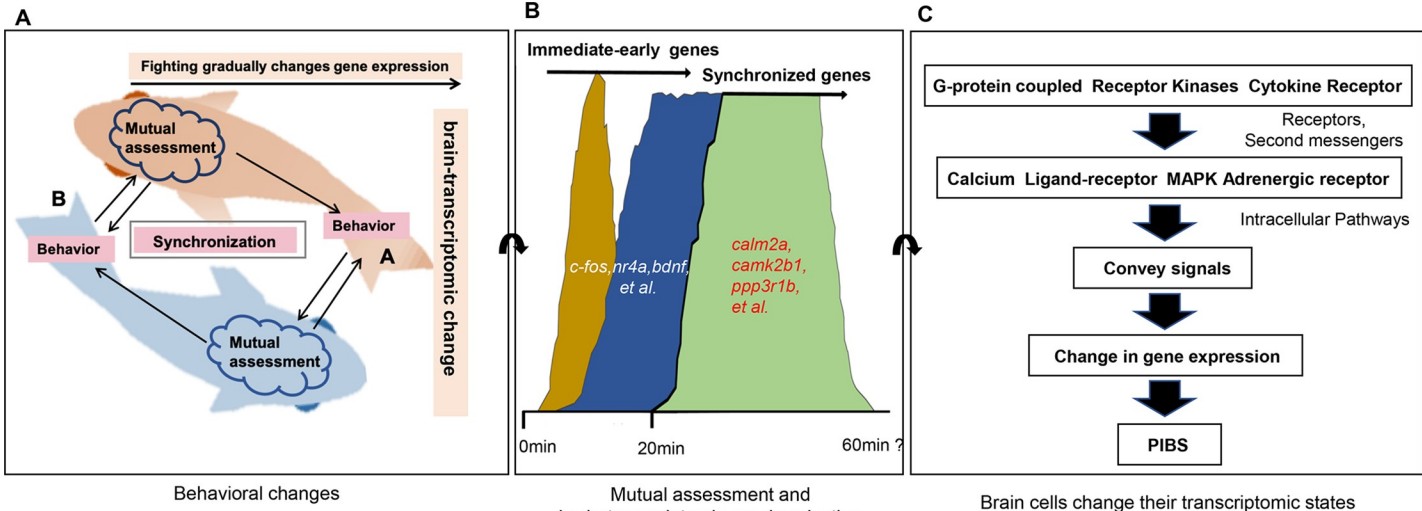

**Fig 7. A model for behavioral and brain-transcriptomic synchronization. (A)** Two fish, A and B, initiate aggressive behaviors (opercular display, fin spread, bite/strike). The repeated attempts of each fighting opponent to match the actions of the other opponent lead to behavioral synchronization. (**B**) Continued physical interaction provides repeated signals for the brains of each fish to adjust the expression of genes in the neurocircuits as mediated by mutual assessment; these genes include immediate-early genes (IEGs), synchronized genes, and learning/memory-associated genes, gradually leading to brain-transcriptomic synchronization. From 0 min to 20 min, different colors are used to refer to transcriptomic plasticity and to partially reflect the gene expression patterns of the DEGs in groups 1 & 2 in Fig 3. From 20 min to 60 min, a single color was used to represent transcriptomic synchronization and to partially reflect the gene expression patterns of the DEGs in groups 3 in Fig 3. (**C**) Learned association through mutual assessment leads to changes in the physiology of the neurocircuits as mediated by second messengers and intracellular signaling pathways to convey signals for brain-transcriptomic synchronization.

formation. Thus, a fighting *B. splendens* seems to activate molecular pathways involved in memory formation during fighting. This observation is further supported by the detected enrichment of the MAPK signaling pathway in the D60-specific gene sets, which is known to be associated with cognitive processes [50]. Several *dual specificity phosphatase* (*dusp*) subunits (*dusp2*, *dusp4*, *dusp3a*, and *dusp7*), modulators of the MAPK signaling pathway [66], were upregulated and synchronized. In particular, the involvement of the calcium signaling pathway appears to be crucial. Different routes of $Ca^{2+}$ entry lead to the activation of different downstream pathways (Fig 6F; S6C Fig and S6D Fig). The activation of genes encoding the NMDAR subunits (*grinaa*, *grin2aa*, and *grin2b*), AMPAR subunit (*grm8a*), and CaV subunit (*cav3*, T-type calcium channels and *cav1*, L-type calcium channels) leads to increases in nuclear $Ca^{2+}$ [67], which in turn binds the *calmodulin* (CALM) subunit (*calm2a*) to activate the *α-calcium calmodulin-dependent protein kinase II* (CAMK2) subunits (*camk1da*, *camk2*, and *camk2b1*) and *calcineurin* (CaN) subunit (*ppp3r1b*), eventually triggering long-term memory formation [68]. Noticeably, each fighting pair activates a unique route within the $Ca^{2+}$ signaling pathway (S14 Table).

Interestingly, the number of up-regulated genes was much higher than that of down-regulated genes among DEGs in both fighting groups (Fig 2C). Our gene enrichment analysis showed that the DEGs in the D20 and D60 were associated with ion transport, synaptic function, and long-term memory. The gene with the highest expression was *thoc2*, a subunit of three prime repair exonuclease (TREX), which is required for efficient export of polyadenylated RNA and spliced mRNA [69]. This asymmetry in gene expression variation between the B and D20/D60 groups was quite striking and is consistent with previous studies in sticklebacks [70], mice [71], and honeybees [72] to some extent. These studies demonstrated that up-regulated DEGs were involved in insulin signaling, oxidative phosphorylation, mitochondrial function, and regulation of metabolic processes, showing deeply conserved metabolic signal

after challenge and representing a shift in the balance of metabolism [71]. Together, highly expressed genes from the D20 and D60 groups suggest the possibility of a shift in metabolism that prefigures later neuronal plasticity in the brain. It is also possible that the asymmetry in DEGs between baseline and fighting groups could be a molecular manifestation of increased inhibitory control of gene expression on aggression so as to avoid an over-escalation [73].

Second, the heatmap and the GoM analyses of the 25 brain samples collected at three different fighting time points together with DEG analysis revealed the presence of brain-transcriptomic synchronization, with more than a half (63.2%) of all DEGs (B vs. D60) involved in this process (Figs 5 and 6). The GoM analysis demonstrated that the composition of multiple clusters from individuals in a pair, especially D60 pairs, was very similar, illustrating nicely the presence of brain-transcriptomic synchronization in a pair-specific manner (PIBS). The GoM model approach has been widely used in population genetics to cluster admixed individuals who have ancestry from multiple populations. Recently, it has been used for assessing global structure in RNA sequencing data in mice and humans [71, 74]. Future studies using this approach along with existing and widely used visualization methods such as LDA and hierarchical clustering (heatmap) will provide a richer summary of the information contained within RNA-seq data [75]. In addition, it will be interesting to find a way to quantify the strength of association between the GoM results and behavioral synchronization of any given pair. Currently, however, as our behavioral analysis here did not include all sets of behaviors requiring synchrony, such as tail-beating, body-orientation, and body-coloration, such quantification will require future studies.

Third, the WGCNA showed that each of the two groups, D20 and D60, was associated with distinct gene modules: MEgrey60, MEblue, and MEturquoise for the D60 group and MEmidnightblue for the D20 group (Fig 4). Although the DEG analysis and WGCNA represent different approaches, they gave consistent results with respect to the characterization of transcriptome changes associated with each fighting group. Of the 564 D60-specific genes identified by the DEG analysis, 491 (87.06%) were present in the three gene modules associated with D60 based on the WGCNA (S6 Table). Our results are consistent with a study that examined DEGs between the winner, loser, and mirror phenotypes (i.e., when a fish fights against its own image in a mirror) in zebrafish using both DEG and WGCNA approaches [6]. Although this study identified fewer gene modules, it showed extensive agreement for the DEG numbers between the two methods [6]. Other recent studies using both DEGs and WGCNA methods have demonstrated fruitful results [71, 72]. Thus, this integrative approach is useful for examining the relationship between gene expression and targeted phenotypes, and one can corroborate the other.

## Transcriptome synchronization using whole brains vs. brain regions

We used the whole brain in this study given that social influences and behavioral changes induce modifications in brain gene expression [76], and that external information can trigger brief variation from the baseline of brain gene expression [77]. Although it is uncertain how these changes are conveyed to other organs, clearly the brain plays a key role in where social experience is integrated with physiological changes to carve the phenotypic plasticity [78]. However, one disadvantage of the analysis using whole brain is that brain tissue consists of a complex mixture of neuronal cell types, each of which has different biological signals, and that some of meaningful signals may be undetected because of their relative low abundance [71,78]. It was found that none of the candidate genes for aggressive behavior in the previous study [28], such as tryptophan hydroxylase (*tph1* and *tph2*), were identified in studies using whole brain such as those in zebrafish [6] and cichlid [27], as well as in the present study. It is

possible that these aggression-associated genes were up- or down-regulated in a specific part of the brain but could not be detected by using whole brain, possibly because any changes were diluted. It has been reported that dominant and subordinate zebrafish showed different number of DEGs in different brain regions after social interaction, such that some genes are overexpressed only in the hypothalamus and hindbrain of the dominant, but other genes are overexpressed in the telencephalon and optic tectum of both dominant and subordinate fish [28]. This suggests that difficulty of analysis for the expression of genes with low abundance in neuronal cells can be overcome by obtaining RNA from smaller area of the brain using micro-dissection or single cell sequencing method.

Brain-transcriptomic synchronization was observed in the analysis of the whole brain, suggesting that several parts of the brain could be involved in the synchronization events observed in this study. In recent years, an approach to map a specialized part of the brain by focusing on regions such as the hippocampus, which is involved in learning and memory [78, 79], has been developed. A comparison of gene expression profiles among brain regions (e.g., telencephalon, diencephalon, cerebellum) in fish responding to intruders [70] has been attempted, and analyses of changes in the conserved 'social behavior network'—a collection of brain nuclei known to regulate social behaviors across vertebrates—based on detailed micro-dissected brain regions in cichlids [80], zebrafish [81], and swordtail fish [82] have been carried out. The limbic region, midbrain, and the cortical region are involved in a variety of motivations in mice [38, 39, 83]. Based on these studies, we propose that the telencephalon region could be a good candidate for further brain-transcriptomic synchronization analysis because it contains structures homologous to the hippocampus and amygdala in mammals and is associated with aggression in other vertebrates [84]. In particular, loci in this region including the dorsal medial (associated with the fear response), dorsal lateral (associated with spatial learning), lateral spectrum (associated with anxiety), pre-optic area (associated with reproduction), and cerebellum (associated with attention) are of interest [85]. A better understanding of the involvement of these specific brain loci in brain-transcriptomic synchronization is a promising area for future research.

## Brain-transcriptomic synchronization reflects cooperative fighting aspect

In the present study, we propose that brain-transcriptomic synchronization might underlie the behavior synchronization between two interacting partners in vertebrates. Furthermore, we speculate that brain-transcriptomic synchronization may occur with collaborative behaviors because of the universality of behavioral synchronization under both competitive and collaborative circumstances, as discussed above.

As the paired fish had very similar behavioral repertoires overall, any transcriptomic commonalities could reflect transcriptomic responses to fighting effort or allocation of that effort rather than being related to behavioral synchrony per se. One possibility for the role of PIBS is that the brain-transcriptomic response in one individual might encode behaviors of its interacting opponent. Previous studies have identified neurons that are selectively activated during a partner's action in songbirds, monkeys, and humans [86–90]. Very recent studies reported that there is inter-brain correlation of neuron activity between two individuals of mice [38] and bats [39] during social interactions. Strikingly, in the case of the mice, correlated brain activity depends on both the encoding of one's own behavior at the cellular level and the behavior of the interacting partner. Moreover, deeper similarities in neural responses were found among friends who view the same movie [91], or between infants and adults while they communicate and play with one another [92]. Also, highly sharing of emotional state within a judgment task was revealed in ravens (*Corvus corax*) [93]. Noticeably, it has been reported that

different neuronal activity patterns induce different sets of activity-regulated genes [94]. Together, the existence of PIBS suggests that correlated brain gene expression may support synchronous neural activity in brains between two interacting partners, reflect cooperative aspects at the molecular level, and indicate the presence of basic forms of empathy in animals [86].

### Brain-transcriptomic synchronization associated with fighting motivation

The neurotransmitter serotonin is involved in regulating various motivations in animals [95]. For example, serotonin plays an important role in prolonging fight duration in lobsters and crayfish [96]. Similarly, expression differences in the serotonin receptor subunits Htr1b and 5-Ht2c account for increased exercise motivation in mice, suggesting that higher motivation for exercise is modulated, at least in part, by the serotonin signaling mediated by the expression of Htr1b and its surrounding chromatin organization [97, 98]. Interestingly, we did find that two additional serotonin receptor subunits, *htr7a and htr7c*, which belong to the neuroactive ligand-receptor pathway were highly expressed and synchronized in four fighting pairs in our data (S14 Table), suggesting the possible involvement of these genes in the persistence of motivation in this system.

In summary, this study demonstrates that fighting behaviors and brain gene expression of the fish *B. splendens* within a fighting pair are highly synchronized. While the behavioral synchronization is observable at an early stage of fighting, the brain-transcriptomic synchronization is discernible after a 20-min fight and is strengthened after a 60-min fight, suggesting the importance of repeated signals for the brain to adjust the expression profiles of certain genes. We demonstrate that brain-transcriptomic synchronization is observed between individuals of a fighting pair in a pair-specific manner after sharing long-term interactions. This is a phenomenon that has been widely known at the behavioral level but has rarely been observed at the molecular level and could occur widely among animals. Altogether, this study provides a framework for an understanding of synchronization at the behavioral and molecular levels that occurs during a contest or collaboration in higher animals through social interactions.

## Methods

### Ethics statement

The animal experimentation procedures used in this study were approved by the Institutional Animal Care and Use Committee (IACUC) (Approval No.106171) of the National Cheng Kung University, Tainan, Taiwan.

### Animals and maintenance

Male fighting fish *B. splendens* used in this experiment were obtained from a local fish shop in Thailand. During delivery, the fish were individually kept in 150-ml glass flasks. In the laboratory, they were transferred to 600-ml glass flasks and housed at $26 \pm 2°C$ with a 12-h dark and 12-h light photoperiod for 2 days for adaptation. They were then transferred to individual tanks in the circulation aquarium system in the laboratory. Fish were fed twice a day with commercial food flakes. The average fish size was $5.2 \pm 1.1$ cm (standard length).

### Behavioral assays

Fish were maintained in the aquarium system tanks for 1 week before a fighting experiment. To test for the interaction, a group of five non-fighting individuals (B1, B2, B3, B4, and B5) was used as the control group. Another 20 adult males, matched on the basis of standard

length, were used for RNA sequencing. Three experimental sets were conducted in which 10 fish (five pairs: D20_11 vs. D20_12, D20_21 vs. D20_22, D20_31 vs. D20_32, D20_41 vs. D20_42, and D20_51 vs. D20_52) were paired for a 20-min fight, 10 fish (five pairs: D60_11 vs. D60_12, D60_21 vs. D60_22, D60_31 vs. D60_32, D60_41 vs. D60_42, and D60_51 vs. D60_52) were paired for a 60-min fight, and 34 fish were paired until the winner chased the loser; this last group was referred to as typical fighting pairs (S1 Table). Fish were individually recognizable based on their coloring, especially the color of their caudal, anal, and dorsal fins. Each pair of fish was introduced into a 1.7-L PVC tank (18 × 12.5 × 7.5 cm). An opaque PVC partition was temporally used to divide the tank into two parts when the two fish were first introduced into the tank to help them become stabilized. After 1 min, the opaque divider was removed. In the first experimental set, the fish were allowed to fight for 20 min; in the second set, the fighting duration was set for 60 min; and in the third set (referred to as the typical fighting pairs), the fish were allowed to fight until the winner chased the loser. Behavioral interactions were videotaped digitally, and the recordings were subsequently used for detailed behavioral analysis. The fish from each experimental set were immediately killed after the fighting interactions with a lethal dose of MS-222 (Syndel USA; 1,000–1,500 mg/L, anesthetic time was ~1 min at 600 ppm concentration), then stored at -80°C for subsequent analysis. The non-fighting group was killed once the fish were taken out of the tanks, and they were removed from the tanks after all fighting group fish were killed (S1C Fig).

The conflicts in all of the D20 and D60 fighting pairs in this study were not resolved. Data obtained from detailed scrutiny of behavioral patterns exhibited during interactions from the 17 typical fighting pairs revealed that a switching point at which the resolution of the fight took place was observed after a 60-min fight. During the 60-min fights, the fighting strength between two opponents within a fighting pair showed almost the same tendency, and the fighting sequences were consistent among all of the fighting pairs. Thus, it is likely that all pairs characterized as "D20" and "D60" were actually in the same stage with respect to fighting sequences.

All 25 brain samples, consisting of five non-fighting samples (B), 10 samples from the 20-min fights (D20), and 10 samples from the 60-min fights (D60), were used to generate RNA-seq data for transcriptomic analysis. The 17 typical fighting pairs were used only for behavioral analysis. With these typical pairs, we referred to the fish by a letter and number to indicate different sets of fighting experiments (e.g., M5 vs. M8 indicates "in set M, fish number 5 fights against fish number 8") and to clarify between the winner and the loser.

## Video analysis

Recorded videos (Nikon Cool Pix E5400) were analyzed using the video editing software Windows Movie Maker (Microsoft). Activities were recorded with respect to both the frequency and timing of biting/striking, surface-breathing, and mouth-locking. Biting/striking was defined as occurring when one fish made open-mouth contact with the other fish. Surface-breathing was defined as occurring when a fish ceased displaying and began gulping air at the surface, and mouth-locking was defined as occurring when one fish took hold of the other fish's upper or lower lip.

## Fighting behavioral structure

To examine the fighting behavioral structure, biting/striking attacks, surface-breathing events, and mouth-locking events were recorded in terms of their frequency and duration. An initial 60-min fight duration was analyzed in which the biting/striking, surface-breathing, and mouth-locking behaviors were broken down into 2-min windows with 1-min overlap for

analysis. To consider the entire 60-min fighting process, the time series was broken up into short, overlapping segments (or time windows). Calculation for frequency and duration values are then determined for each of these time windows (1 min in this case). The resultant time sequence of all frequency and duration values from each time window can indicate behavioral changes over time. The length of the time window was determined based on the type of movement considered, along with the observed time scale for behavioral changes [99]. Pearson correlation coefficients for bite/strike and surface-breathing (S2 Table and S3 Table) were separately calculated between paired fish using the *cor* function in R/Bioconductor. The figure panels showing the combination of all behaviors (Fig 1C and 1D, S2 Fig) were generated using Excel.

## Tissue processing and RNA extraction

Brains were rapidly dissected from the three groups of frozen fish (B, D20, and D60) in PBS buffer and were individually collected in 1.5-ml tubes containing 1,000 μl Trizol for further processing. Total RNA from each whole brain (0.01 g/brain) was extracted using Trizol and then purified with Quick-RNA MiniPrep (Zymo Research, USA). The concentration and purity of the RNA was determined by Qubit (Eugene, Oregon, USA) and a BioAnalyzer 2100 RNA Nano kit (Agilent, USA), respectively. The RNA quality integrity (RIN) values of the samples ranged from 6.3 to 8.8. Total extracted RNA was kept at –80˚C until processing.

## Stranded RNA sequencing

RNA-seq libraries were constructed using the TruSeq Stranded mRNA Library Prep Kit (Illumina, USA) with proper quality control, and the molar concentrations were normalized using a KAPA Library Quantification Kit (Kapa Biosystems, USA). The 25 RNA libraries were sequenced in two separate groups. Five RNA libraries (B1, B2, B3, D60_11, and D60_12) were sequenced on the Illumina HiSeq 2500 system at Yourgene Bioscience Co., Ltd. (Taipei, Taiwan) using paired-end sequencing. The other 20 samples (B4, B5, D20_11, D20_12, D20_21, D20_22, D20_31, D20_32, D20_41, D20_42, D20_51, D20_52, D20_61, D20_62, D60_21, D60_22, D60_41, D60_42, D60_51, and D60_52) were sequenced on the Illumina HiSeq 2500 system at the NGS High Throughput Genomics Core (Biodiversity Research Center, Academia Sinica, Taiwan) using single-end sequencing. The read lengths and total sequencing reads are shown in S4 Table.

Given that the samples had undergone two different sequencing methods, we examined whether the paired-end vs. single-end difference led to any biases in the data using a multidimensional scaling (MDS) plot. The MDS plot, which was color-coded based on the sequencing method, revealed that the two methods resulted in slight or no biases as all samples were clustered together (S7 Fig). We further evaluated the extent to which the two methods yielded equivalent results. We used only the forward-reads for the pair-end sequencing samples and analyzed all data in single-end mode throughout the pipeline to minimize any bias. The results obtained were comparable in terms of mapping rates and the total number of genes across all the samples (S4 Table) between the two methods, which is consistent with a previous study [100] and indicated sufficient quality for further downstream analyses.

## Quality processing and read mapping

Adaptors were trimmed using the Cutadapt tool [101] with the following command line, -b GATCGGAAGAGCACACGTCTGAACTCCAGTCAC -b AGATCGGAAGAGCGTCGTG-TAGGGAAAGAGTGT -o $1.output file name $1.inputfile name. The low-quality bases and reads were removed by using the fastq_quality_trimmer tool version 0.0.13 (http://hannonlab.

cshl.edu/fastx_toolkit/) with the following parameters: fastq_quality_trimmer -t 20 -l 30 -Q 33 -i $1.input file name | fastq_quality_filter -q 20 -p 80 -Q 33 -o $l.output file name.

The *B. splendens* genome and its gene IDs were downloaded from http://gigadb.org [32]. The processed sequencing reads were mapped to the genome using Tophat version 2.1.1 [102], and its embedded aligner Bowtie2 version 2.1.0 with the default parameters [103]. The unique mapping reads (reads that matched the reference genome at only one position) were extracted using samtools with the following command line view, -q 4 $1.input file name >$1.output file name. The exon-mapped reads were counted by featureCounts [104]. The normalized expression levels of genes, represented by the trimmed mean of M-values (TMM) [105], were generated by the "edgeR" package (version 3.26.8) in R [106]. In total, 23,306 genes from each brain sample were generated. All raw data are available at Bioinformation and DDBJ Center (https://www.ddbj.nig.ac.jp/index-e.html), accession code number DRA009599.

### Identification of DEGs

The DEGs were obtained from comparisons of gene expression between the non-fighting group (B) with the D20, and B with the D60. We included genes with at least one count per million in at least one sample. The p-values from all contrasts were adjusted via empirical FDR at once. Cut-off values of FDR ≤ 0.05 and |log FC| > 2 were used to select the DEGs (S6 Table; source code, edgeR).

### Brain-transcriptomic synchronization analysis

To evaluate the transcriptomic synchronization, we first calculated Pearson correlation coefficients of $\log_2$-transformed TMM values (r) separately between two random fish from each group of the D20 and D60 using the *pairs* function in R. Combinations of all possibilities between two random fish among the 10 fish resulted in a total of 45 pairwise (five paired fish and 40 unpaired fish) for each fighting group (45 = 2-permutations of 10; S4 Fig). Next, to assess how similar the transcriptomes of paired fish were and to what extent the paired fish were synchronized, we compared the r values from paired fish to the r values from unpaired fish from the same treatment (e.g., D20_paired fish vs. D20_unpaired fish; D60_paired fish vs. D60_unpaired fish) using a permutation test, which is implemented by the "exactRankTests" package in R with all default settings. The hypergeometric test (23,306 total genes analyzed) was conducted by *phyper* function in the "dplyr package" in R.

### Clustering

To differentiate transcriptomic responses among the three groups (B, D20, and D60), a principal component analysis (PCA) and a linear discriminant analysis (LDA) were done on the same data set of the top 50% most variable gene transcripts (11,653 genes sorted by standard deviation) from the entire gene set after undergoing DEG analysis in edgeR (n = 23,306 genes). Both the PCA and LDA analyses were implemented using the "MASS" package in R (S4E Fig; source code, LDA). To visualize transcriptomic synchronization, a grade of membership (GoM) analysis was performed using the entire gene set (n = 23,306 genes) with '7' as the selected number of clusters based on the "CountClust" package in R [75] (source code, GoM). To cluster gene expression patterns of each fighting group and each individual, heatmaps were created using the pheatmap (Fig 3C) and ggplot2 (Fig 5B) functions in R. The consensus tree and bootstrap values for each tree node in Fig 3C and Fig 5B were obtained with the *hclust* function in the "pvclust" package in R/Bioconductor. A

Venn diagram was constructed using webtools (http://bioinformatics.psb.ugent.be/webtools/Venn/).

## WGCNA

Using normalized expression counts from all the genes that underwent differential expression analysis in edgeR (23,306 genes, described above), we characterized gene expression network dynamics. This analysis was performed using an open-source software, WGCNA package in R [74]. WGCNA shows patterns of co-expressed genes (modules, which are arbitrarily assigned a color name by the software package) based on the eigengene value (a variable derived from the first principal component of expression in a module that represents the gene expression profile in a module, see [74] for more details). Our goal was to identify modules associated with the fighting groups (B, D20, and D60). We calculated Pearson correlation coefficients between all gene pairs in all three fighting groups from the experiment: B, D20, and D60. Further, we used DAVID as described below to perform GO and KEGG pathway analyses on the chosen gene modules in WGCNA. The designed matrix for the WGCNA used to identify gene modules is shown in S15 Table. We used the default parameters for all WGCNA settings (source code, WGCNA).

## Characterizing the synchronized gene set

We characterized the synchronized gene set with the following two-step method.

In the first step, the DEGs from the B vs. D20 and B vs. D60 comparisons (FDR < 0.05 and log FC > 0) were obtained as described above.

In the second step, we divided the procedure into three parts as follows.

(1) To determine whether a gene i is synchronized or not, the expression distance for each particular gene between two random fish among the total of 45 possible combinations (45 = 2-permutations of 10, five paired fish and 40 unpaired fish) was separately computed for each group (D20 and D60) using the following formula:

$$D_{ij} = |\log_{10}(A_{ij}/B_{ij})|$$

where $D_{ij}$ is the distance for gene i between a random pair j (A and B) and $A_{ij}$ and $B_{ij}$ are the TMM values for gene i of fish A and fish B from the pair j, respectively.

(2) A permutation test for each Di between the five paired fish and the 40 unpaired fish was separately conducted for each of the fighting groups, D20 (D20-paired fish vs. D20-unpaired fish) and D60 (D60-paired fish vs. D60-unpaired fish), to assess the statistical significance. The tests were implemented by the "exactRankTests" package in R with all default settings. A value of p < 0.05 was considered indicative of a synchronized gene.

(3) To determine in which fighting pair(s) gene i is synchronized, we first separately computed the expression distance for each particular gene i between two unpaired fish for each group (D20 and D60) using the following formula:

$$D_{up} = |\log_{10}(C_{ik}/D_{ik})|$$

where $D_{up}$ is the distance for a gene i between an unpaired fish k (C and D); $C_{ik}$ and $D_{ik}$ are the TMM values for a gene i of fish C and D from the pair k, respectively.

Then, a baseline for each synchronized gene ($B_i$) was identified by calculating the 25th percentile value for the distance of each particular gene (i) between the 40 unpaired fish for D20 and D60 separately (see Fig 6D, left). Finally, we compared the distance of synchronized gene i in pair j ($D_{ij}$) with the baseline ($B_i$). If $D_{ij} < B_i$, then the synchronized gene i is defined as synchronized in the pair j.

## Obtain learning and memory genes

To find genes related to "memory" or "learning", the sequences from the 1,522 synchronized genes in the D60 group were searched against the amino acid sequences related to "memory" or "learning" obtained from the UniProt database (https://www.uniprot.org/) using blastx with a cut-off E-value of 1e-20.

## Gene annotation

The protein sequence table of *B. splendens* was obtained from http://gigadb.org/dataset/100433. These sequences were used for a BLAST analysis against protein databases to find the homologs of zebrafish (*Danio rerio*) using Blast2GO (https://www.blast2go.com/blast2go-pro). As a result, we obtained 15,754 well-annotated genes for the GO and KEGG pathways analyses.

## Functional gene enrichment analysis

The significantly enriched Gene Ontology terms (biological process) and KEGG pathways were identified by the Database for Annotation Visualization and Integrated Discovery (DAVID) [107]. A p-value cut-off of $< 0.05$ was used to select for significantly enriched functional terms. The GO terms obtained from DAVID were summarized into larger and more general categories to get a general overview of the underlying biology using REVIGO [108]. Terms were grouped together if they were in a similar pathway and/or based on semantic similarity. GO enrichments along with their respective p-values are in S10–S12 Tables

# Supporting information

**S1 Video. Fighting between two males of *B. splendens*.**
(MP4)

**S1 Fig. Dynamic fighting behaviors of two male *B. splendens* individuals. (A),** Two male *B. splendens*. **(B),** Sequence of fighting behaviors that take place in order from (1) to (6) throughout the fighting process. (**C**), Sample collection. We conducted two fighting experiments (n) per day beginning at 1 PM ($t_0$), and fish were immediately sacrificed at specific time points ($t_1$) by submersion in the lethal dose of MS 222. It took 1 day to collect the five individuals for Set 1, 3 days to collect the five pairs for Set 2, followed by another 3 days to collect the five pairs for Set 3 and another 3 days for Set 4. After sacrifice, the samples were immediately transferred to a -80˚C freezer and were stored there until subsequent brain dissection, RNA extraction and RNA sequencing.
(TIF)

**S2 Fig. Behavioral differences among all fighting pairs. (A-C, G, H)** Behavioral differences among all fighting pairs with respect to biting/striking. **(D-F, I, J)** Behavioral differences among all fighting pairs with respect to surface-breathing. These are data from five typical fighting pairs (Set 4 in S1C Fig) that were not included in the D20 or D60 group.
(TIF)

**S3 Fig. Gene Ontology and KEGG pathways. (A-E)** Bar charts showing the enriched biological processes associated with four gene modules generated from the WGCNA analysis—MEmidnightblue (A), MEgrey60 (B), MEblue (C) and MEturquoise (D)—and the MAPK signaling pathway components associated with three of these modules (E). **(F-I)** Significantly enriched KEGG pathways for those modules associated with the fighting groups: MEgrey60 (G), MEblue (H) and MEturquoise (I) for the D60 and MEmidnightblue (F) for the D20; up-

regulated genes are shown in red.
(TIF)

**S4 Fig. Correlation coefficient values (r), representative synchronized genes, WGCNA, and PCA. (A)** The r values between the TMMs for the 23,306 gene transcripts from the two opponents of the fighting pairs in the D60 group; **(B)** The r values between the TMMs for the 23,306 gene transcripts from the two opponents of the fighting pairs in the D20 group. The r values in red boxes are for the fighting pairs. Bivariate scatter plots are shown below the diagonal, histograms are shown on the diagonal, and the Pearson correlation values are shown above the diagonal. **(C)** Representative genes showing pair-specific synchronization of expression. Although these genes were synchronized in all five pairs, the level of synchrony differed for each particular pair. **(D)** The WGCNA heatmap showing values for r (upper value) and p (lower value in parentheses) in all of the gene modules. **(E)** Clustering of all 25 brain samples using a principal component analysis (PCA); blue, non-fighting group (B); red, D20 group; and green, D60 group.
(TIF)

**S5 Fig. Venn diagram for synchronized genes and heatmap for all gene set. (A)** Venn diagram generated from the 1,522 synchronized genes from five pairs of the D60 group (D60_11 vs. D60_12, D60_21 vs. D60_22, D60_31 vs. D60_32, D60_41 vs. D60_42, and D60_51 vs. D60_52); **(B)** Venn diagram generated from the 172 synchronized genes from five pairs of the D20 group (D20_11 vs. D20_12, D20_21 vs. D20_22, D20_31 vs. D20_32, D20_41 vs. D20_42, and D20_51 vs. D20_52). **(C)** Heatmap using the 25 cDNA libraries with each of the 23,306 gene contigs (all gene set). Intensity of color indicates expression level (red, high expression; blue, low expression). Similarities between individuals within fighting pairs and between fighting pairs as shown by hierarchical clustering can be seen above the heatmap. Bootstrap values at the nodes were obtained by *hclust* function in R.
(TIF)

**S6 Fig. Brain-transcriptomic Synchronization Associated with Long-Term MemoryGenes. (A-C)** Calcium pathways generated from the synchronized gene set in the D60 group and non-synchronized gene sets of the D20 and D60 group **(B, C). (D)** *Glutamate* (Glu) is released from the presynaptic terminal and acts on both postsynaptic *N-methyl-D-aspartate receptors* (NMDARs; *grinaa, grin2aa, grin2b* subunits) and *α-amino-3-hydroxy-5-methyl-4-isoxazolepropionic acid receptors (*AMPARs; *grm8a* subunit) to depolarize these receptors and release $Mg^{2+}$. **(E)** $Ca^{2+}$ increases in the postsynaptic terminal and binds to *Calmodulin* (CaM; *calm2a* subunit) to activate the *calmodulin-dependent protein kinase (*CAMKII; *camk1da, camk2a, camk2b1* subunits). Also, *β-adrenergic receptors* (β-ARs) are used to recruit more AMPAR to promote long term memory.
(TIF)

**S7 Fig. The MDS plot color coded by sequencing methods.** B, non-fighting group; D20, the D20 fighting group; D60, the D60 fighting group. Red circle clusters the paired-end sequencing and blue circle clusters the singe-end sequencing.
(TIF)

**S1 Table. Seventeen typical fighting pairs and those parameters used to construct the representative schematic fighting structure.**
(XLSX)

**S2 Table. Correlation coefficients of biting and striking attacks for eight typical fighting pairs used to visualize behavioral synchronization.**
(XLSX)

**S3 Table. Correlation coefficients of surface-breathing events for eight typical fighting pairs used to visualize behavioral synchronization.**
(XLSX)

**S4 Table. The mapping rate of 25 *B. splendens* brain samples from the non-fighting, D20, and D60 groups to the reference *B. splendens* genome.**
(XLSX)

**S5 Table. Differential Expression Genes List (DEGs) (average TMM value).**
(XLSX)

**S6 Table. Number of genes in each module from the WGCNA analysis.**
(XLSX)

**S7 Table. Top 5 driving genes in each cluster extracted from the Grade of Membership (GoM) analysis.**
(XLSX)

**S8 Table. List of DEGs generated from a comparison of the expression levels from the non-fighting group vs. the D60 group (FDR $\leq$ 0.05).**
(XLSX)

**S9 Table. List of synchronized genes expressed in one to five pairs of fighting *B. splendens*.**
(XLSX)

**S10 Table. GO terms among the synchronized genes and non-synchronized genes in the D60 group with respect to the biological process (BP) list (p < 0.05).**
(XLSX)

**S11 Table. GO terms among the synchronized genes and non-synchronized genes in the D60 group with respect to the molecular function (MF) list (p < 0.05).**
(XLSX)

**S12 Table. GO terms among the synchronized genes and non-synchronized genes in the D60 group with respect to the cellular component (CC) list (p < 0.05).**
(XLSX)

**S13 Table. KEGG analysis using synchronized genes related to learning or memory and for which there were at least two genes in the pathway (p < 0.05).**
(XLSX)

**S14 Table. Commonly enriched pathways and their constitutive components as represented among the synchronized genes in the D60 group (p < 0.05).**
(XLSX)

**S15 Table. Behavioral matrix for the WGCNA analysis.**
(XLSX)

## Acknowledgments

This study was completed at the Foundation for Advancement of International Science (FAIS) and Kitasato University, Japan. We thank Mr. Y. Otake, a president of FAIS, and Mr. T.

Sugino, a president of ZENICK Corp., for support. We thank Drs. P-W. Gean and K. Yamaguchi for discussions and the preliminary analysis of the data. We are also grateful to Drs. K. Honma, J. Wang and S. Nishimura for critical reading of the manuscript, to Ms. A. Koizumi for improving the quality of the figures, and to Mr. K. Onodera at FAIS for assistance. We thank Drs. Alison Bell (Guest Editor), Gregory Barsh (Editor-in-Chief), and the three anonymous reviewers for their constructive comments, which have improved the manuscript.

## Author Contributions

**Conceptualization:** Norihiro Okada.

**Data curation:** Trieu-Duc Vu, Yuki Iwasaki, Shuji Shigenobu, Norihiro Okada.

**Formal analysis:** Trieu-Duc Vu, Yuki Iwasaki, Shuji Shigenobu, Kenshiro Oshima, Chao-Li Huang, Takashi Abe, Satoshi Tamaki, Yi-Wen Lin, Masaru Hojo, Hao-Ven Wang, Shun-Fen Tzeng, Hao-Jen Huang, Akio Kanai, Norihiro Okada.

**Funding acquisition:** Norihiro Okada.

**Investigation:** Norihiro Okada.

**Methodology:** Shuji Shigenobu, Norihiro Okada.

**Project administration:** Norihiro Okada.

**Supervision:** Takashi Gojobori, Tzen-Yuh Chiang, H. Sunny Sun, Norihiro Okada.

**Validation:** Trieu-Duc Vu, Yuki Iwasaki, Norihiro Okada.

**Writing – original draft:** Trieu-Duc Vu, Norihiro Okada.

**Writing – review & editing:** Trieu-Duc Vu, Yuki Iwasaki, Akiko Maruko, Kenshiro Oshima, Erica Iioka, Chao-Li Huang, Chih-Kuan Chen, Mei-Yeh Lu, Wen-Hsiung Li, Norihiro Okada.

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
