## [Decision Letter · Decision Letter 0]

15 Jan 2020

Dear Dr Okada,

Thank you very much for submitting your Research Article entitled 'Behavioral and neuro transcriptomic synchronization between the two opponents of a fighting pair of the fish Betta splendens' to PLOS Genetics. Your manuscript was fully evaluated at the editorial level and by independent peer reviewers. The reviewers appreciated the attention to an important problem, but raised some substantial concerns about the current manuscript. Based on the reviews, we will not be able to accept this version of the manuscript, but we would be willing to review again a much-revised version. We cannot, of course, promise publication at that time.

If you decide to revise the manuscript for further consideration at PLOS Genetics, please aim to resubmit within the next 60 days, unless it will take extra time to address the concerns of the reviewers, in which case we would appreciate an expected resubmission date by email to plosgenetics@plos.org.

[LINK]

We are sorry that we cannot be more positive about your manuscript at this stage. Please do not hesitate to contact us if you have any concerns or questions.

Yours sincerely,

Alison Bell

Guest Editor

PLOS Genetics

Gregory Barsh

Editor-in-Chief

PLOS Genetics

Comments from Guest Editor:

The paper by Yu et al reports the results of a study of synchronization of behavior and brain gene expression in Betta splendens. The authors staged fights between two individual male fish, recorded their behavior, and sacrificed them 20 or 60 minutes after the fight. The authors report behavioral synchrony between individuals within a pair, as well as evidence for brain gene expression synchrony. The authors frame their paper in terms of gaining insights into the molecular mechanisms of assessment, which has been well studied from a behavioral and ecological point of view, but the underlying mechanisms are unknown.

The paper was reviewed by three expert reviewers. All three reviewers were generally positive about the overall study and motivation but they raised a number of substantive concerns about the presentation of the study, the rationale, the analysis and the interpretation of the results.

For example, reviewer #1 pointed out that the paper in its present form does not clarify why brain gene expression (as opposed to, say, neural activation) was measured. I strongly agree with this comment – why do the authors think that brain gene expression is operating over a time scale relevant to behavioral synchrony during a fight? The reviewer also pointed out that at present the study presents behavioral and brain gene expression data with some sophisticated analyses but there are no follow-up studies presented, nor attempts at validation. Over the past several years the bar for publishing pure gene expression papers has been raised – are the brain gene expression results sufficiently insightful to merit publication without additional studies of, e.g. neural activation, manipulation of candidate pathways, etc? Reviewer #1 also noted that it wasn’t clear if the fights were resolved in any of the pairs – this is a critical detail that should be clarified.

Reviewer #2 pointed out several weaknesses of the study that question the biological significance of the findings. Reviewer #2 suggested several different statistical approaches that could strengthen confidence in the findings, e.g. by considering the effect of pair as a random effect, using a two-way ANOVA rather than separate pair wise tests. The authors should also consider Reviewer #2’s point about whether covariance structure was the same among the different treatment groups – this is assumed by the WGCNA analysis and unequal covarainces could cause spurious findings. Reviewer #2 also had some really good ideas for ways to rigorously evaluate evidence for greater synchrony among individuals within versus between pairs, e.g. via permutation. Reviewer #2 also brought up some alternative interpretations of similarity between individuals within a pair, e.g. they reflect similar fighting effort rather than synchrony per se.

Reviewer #3 was generally enthusiastic about the paper and made a number of helpful suggestions, e.g. justification of the whole brain analysis, and an explicit statement about which brain regions might be good candidates for future study. Reviewer #3 questioned the utility of coining the term “PINS”. This is something the authors should consider. Reviewer #3 also questioned whether the TFBS analysis merits inclusion in the main body of the paper given that the results do not add much to the overall message of the study.

All three reviewers raised a number of other helpful points that the authors should find useful when revising their paper.

Overall I agree with all three reviewers and think that this study has the potential to make a valuable contribution to the literature but it needs work. The claims need further substantiation with more detailed and rigorous analyses, and the authors should seriously consider whether they had sufficient statistical power and quantitative resolution (e.g. whole brain) to make some of these claims. It would be more convincing if it was clear that the authors tried many tactics to reject their hypothesis.

Reviewer's Responses to Questions

**Comments to the Authors:**

Reviewer #1: This ms by Vu et al. aims to uncover the changes in whole brain transcriptome in a pair of fighting males that engage in particular mouth locking behaviour. The authors use Betta splendens fish as a model to create fighting pairs. They quantify behaviour and gene expression by RNA-seq in fighting males that are in a fight for 20 minutes and 60 minutes and controls that are isolated. They find that in the later time point, gene expression is correlated between the two males of a pair (compared to controls or compared to a fighting male from another pair).

As a reader, it is crucial for me to clearly see in a ms what we know on the topic and what we don’t, and why its important to know more about this. In the ms I did not see a clear reason for studying gene expression rather than neural activation. Telling the reader what the ms will bring is essential. I was also surprised that there were no clear hypothesis nor predictions. I have several questions about the methods.

Specific comments

Introduction

After line 100, it wold be essential to explain why the authors want to learn about the neurotranscriptomic state of males during a fight. Why should we study this, what will it bring to our understanding of behaviour in general. This is important for the broad audience that is targeted by this journal. It would be essential to have hypotheses and predictions, at least proposing alternative models of what should happen, and in which type of genes, given the behavioural data that is available and what we know from studies in males that have already resolved conflicts (dominant/subordinate for example).

Line 106-108 and results line 150-153

For the whole ms, it is not clear if conflicts were resolved or not in the D20 and D60 fish. If not, can an index of escalation or of when in the timeline the pairs are towards resolution (creating an index based on the “typical fighting pairs” maybe?) be created? This analysis looks like a time series analysis, but we are not sure if each pair characterised as “D20” and “D60” is actually in the same stage. We they always killed while mouth locking? Etc?

Results

Line 154

Please explain why the top 10 000 highly expressed genes were used. And what does highly expressed mean? Highest read count? What does it mean biologically?

Line 178-179

Why discuss the fact that some genes have a particular biological function? Is this the result of an enrichment analysis or just cherry picking genes?

Line 181

How can duplicates exist? What do they represent? Would it be better to remove them before the statistical analysis of DEG?

Line 195

The WGCNA is not explained in the methods and we have no idea how it answers the question about the neurotranscriptomic state of fighters.

Line 200

Please explain how a gene module eigen value can be correlated with 3 treatments. How can one correlate continuous values with 3 discrete groups? What can of correlation can be used if one of the variable has no value / cannot be ordered?

Line 218-224

What is the meaning of these results? What question do they answer?

Line 317-337

This analysis really looks circular to me. The authors need to explain why extracting genes based on their function (learning and memory) are then used in a kegg analysis to find enriched pathways (?) and one of these pathways but not the others (why?) related to calcium signalling pathway is then used to analyse the gene sets (including the one from which the genes related to memory where extracted from in the first place!) to determine if each gene set is enriched for genes related to learning. Please explain how this is not circular.

METHODS

Line 604

Why use a cut off of FC > 2?

Line 606-607

If 45 pairwise correlations are done, one needs to correct for multiple testing?

Line 610-611

Please give more detail for the LDA and GoM, presently they are impossible to replicate

Line 612-614

The heat map combined a k-mean clustering and hierarchical clustering according to the figure caption. Please give detail on the parameters used for the hierarchical clustering (distance measure, model for clusters) and for the k-means (how were the number of k clusters chosen?). Also, how where the TMM data normalised?

Line 651

The linux pipeline should be on github or another repository for real reproducibility rather than available on request

Figures

Figure 2B is of poor quality

Figure 3A the venn diagram is for all genes, no matter in which direction they changed expression? Are they always in the same direction in D20 and D60 compared to controls?

Minor comment

Line 332

Here it says log FC > 2, probably mean log FC > 0 ?

Line 582

The authors refer to sample D50_21 and D50_22 but I suppose they mean D60?

Reviewer #2: This manuscript analyzes behavior and gene expression changes in fighting fish. The manuscript is generally well-written and includes sophisticated analyses to address novel questions. My enthusiasm for the paper is reduced by several analysis issues that leave me wondering how many of the findings are trivial consequences of the analyses vs. findings of true biological significance.

General concerns about behavioral synchronization findings:

(1) To what extent is the calculated synchronization a trivial consequence of the inherent synchrony of mouth locking + the reality that other behaviors occur in a defined order vs. a more active form of synchronization? Is there a way to evaluate the expected level of synchrony that takes into account mouth-locking as a constraint?

(2) The methods mention a 1-minute overlap for analysis, but I was not sure what this meant. Please explain more fully the justification for the window approach.

General concerns about DEG analyses:

(1) Given the experimental design, I would recommend an ANOVA rather than two pairwise tests. ANOVA would more appropriately take into account the use of the same control group for the D20 and D60 groups & would allow a formal assessment of differences between D20 and D60 directly rather than making inferences based on the lack of or presence of overlap in separate DE analyses. Also, I suggest those models take into account the non-independent nature of the paired fish (e.g. a random effect for pair).

(2) I read Figure 2A quite differently than the statements on lines 155 – 158. As I see it, Axis 1 separates D20 and control from D60, and Axis 2 separates D20 and D60 from controls.

General concerns about WGCNA:

(1) I find it highly likely that the three groups differ in the co-expression structure (especially the control group compared to the fighters, but perhaps D20 and D60 as well). I therefore would not combine all the animals into one WGCNA analysis, as the differential expression between groups should drive the correlations identified and lead directly to findings such as close associations between modules and DEGs as reported in lines 218-219 and discussed in 459-470.

(2) Line 200 states that the associations between the eigengene and fighting groups were computed using correlations. I am not sure based on this phrasing what analyses were conducted, but I don’t understand how correlations could be applied in this context.

General concern about transcriptomic synchronization:

My largest concern with the paper is the transcriptomic synchronization analysis (I adopt the authors’ terminology here, but see my suggestion about this word choice below). The analyses assess whether the transcriptomes of two paired fish have more in common than the transcriptomes of two random fish from unknown behavioral state (so fish that fight have more in common than noise, basically). I’d expect any two fish in the D20 or D60 group – not just those that fought each other – to appear synchronized by this measure. If the manuscript wants to assess how similar the transcriptomes of paired fish are, I recommend revising the analysis to compare the paired fish to unpaired fish from that same treatment, omitting the comparison to the non-fighting fish. Figure S4 does suggest this would be a promising approach, but the t-test of the correlation coefficients presented in Figure S4 is not appropriate because the individual pairwise correlations are not independent of one another. Permutation/resampling approaches would offer appropriate alternatives to make general comparisons as to how much paired fish were synchronized as a first step to warrant follow-up characterization of PINS (e.g. permuting which fish are paired then comparing a metric summarizing all correlations of the two transcriptomes, not the correlations in each gene individually). To identify the PINS, I would suggest designing another permutation or resampling test to set a threshold for which genes are PINS rather than relying on a t-test – both correcting for multiple hypotheses and also not requiring the data meet the assumptions of the t test. Because of this fundamental disagreement with the methods used to identify PINS, I have not commented on the broader discussion of the PINS themselves.

If a revised analysis as proposed found that transcriptomes for paired fish were more synchronized than fish selected at random, I suggest that the authors consider all possible cause and effect relationships, as the paired fish had very similar behavioral repertoires overall so any transcriptomic commonalities could reflect transcriptomic responses to fighting effort or allocation of that effort rather than being related to behavioral synchrony per se (considering that the behaviors were measured quite a bit before the mRNA expression, and the mRNAs would typically influence physiology only after protein synthesis and trafficking).

Specific comments:

Line 57-58: I don’t see the link between data presented and facial convergence or empathy.

Line 178-180: State method/evidence for this statement.

Line 433-440: References needed for these statements.

Line 445-447: The conclusion as stated in this sentence oversteps the evidence presented.

Line 544-547: How long does this overdose with MS222 take? The main concern would be the extent to which gene expression might change during the euthanasia.

Line 582: I wasn’t sure if D50_21 and D50_22 was an error in naming or whether these fish had only 50 minutes of fighting?

Line 578-585: Given that the samples had two difference sequencing methods, I recommend adding a sentence that summarizes the extent to which the methods yielded equivalent results and addressing whether the paired vs single end difference led to any biases in the data. Some sort of MDS or PCA plot color coded by sequencing run might also help assuage concerns.

Figure 2B has D2_31 listed twice.

General stylistic suggestions:

Although ‘neurotranscriptomic’ and similar terms are used in the literature, I think the term ‘transcriptomic’ is preferable given that brain samples include transcripts from many non-neuronal cells that also have dynamic responses to stimuli and may play important functional roles.

Also, aside from the major concern about the data analysis supporting the transcriptomic synchronization, a terminology suggestion is that the word synchronization implies coordinated timing. The transcriptomic analyses might reflect coordination, but I would avoid the term synchronization because we don’t have temporal dynamics here.

Reviewer #3: In the manuscript “Behavioral and neurotranscriptomic synchronization between the two opponents of a fighting pair of the fish Betta splendens,” Trieu-Duc et al. analyze the brain transcriptomes of Siamese fighting fish at 20 min and 60 min after the initiation of combat. They find a considerable number of differentially expressed genes, compared to a control group, as well as a remarkable amount of co-variance in gene expression in fighting pairs, especially at the 60 min time point. The authors suggest several explanations for this putative “synchronization.” Overall, this is a well-executed study with many interesting and novel results. Below, I list several general and specific comments, which hopefully will help to improve the manuscript further.

General comments:

1) Behavioral and physiological synchronization have long been studied by scientists. Although the authors cite a few studies in the Discussion, their treatment of this phenomenon is superficial, given its importance in reproductive/mating behavior, cooperative behavior, aggressive behavior etc. I suggest the topic is best introduced in the Introduction. Further, the recent papers from the labs of Weizhe Hong and Michael Yartsev (which the authors cite later) provide excellent physiological evidence for synchronization. Decades of behavioral evidence as well as these recent physiological findings suggest synchronization at the gene expression levels as a hypothesis.

2) The authors examined whole-brain transcriptomes. It would be good to provide a rationale for this decision and discuss its limitations. Then, in the Discussion, when pointing at future experiments, consider which brain regions might be good candidates for further analysis.

3) What is the rationale for selecting the 10,000 most highly expressed genes in the LDA? Why not instead use the 10,000 (or 50%, or similar) most variable genes?

4) The asymmetry in gene expression variation between baseline controls and D20 / D60 animals is quite striking (Figure 2C/D). How should we interpret this strong bias towards more highly expressed genes in the fighting animals? How does this result compare to biased transcriptomes described previously in other systems? Also, consider showing volcano plots for both B vs D20 and B vs D60, and consider volcano plots where the D20 DEGs are projected onto the B vs D60 volcano plots and vice versa.

5) I am not convinced there is good reason to introduce a new acronym “PINS.” What the authors are showing is increased covariance in gene expression between combatants compared to baseline controls, especially at the 60 min mark. They interpret this covariance as “synchronization,” which is fine but should be limited to the Discussion. “PINS” seems a little vacuous and excessively speculative.

6) Do the observed (synchronized) gene expression patterns give insight into the neuromolecular mechanisms underlying sequential assessment, as laid out by Enquist & Leimar sand many others since then?

7) The survey of DEG promoters in search of overrepresented transcription factor binding sites is superficial and distracting. It does not add much to the study, so I suggest either to remove it entirely or to move it to supplemental materials.

Specific comments:

8) What was the rationale for choosing the 20 min and 60 min time points, respectively?

9) Figure 3C: This heatmap is fine, though the legend is illegible. I would like to see hierarchical clustering of all samples for, say, the top 50% most variable genes. How clearly are the three groups (B, D20, D60) separated? Do paired individuals cluster together? I believe the heatmap in Figure 5B attempts this (though why is it part of Figure 5?), but has way too many uninformative (i.e., barely variable) genes and the dendrogram has not been bootstrapped. Like all other clustered heatmaps in this study, dendrogram should have bootstrap values (consider using R package pvclust).

10) What is the co-variance structure between module eigengenes and various behavioral metrics?

11) The GoM analysis is very cool and convincing! Are there any test statistics that would allow us to estimate the strength of the association? Is there a way to quantify the strength of association between GoM and behavioral synchronization of any given pair?

12) While much talked about, the “mirror neuron” concept is quite contentious among systems neuroscientists, with good reason to think that they are an artefact. I strongly suggest removing this discussion.

**Have all data underlying the figures and results presented in the manuscript been provided?**

Reviewer #1: Yes

Reviewer #2: No: I didn't see mention of where raw sequencing data will be available, but perhaps I missed it?

Reviewer #3: Yes

PLOS authors have the option to publish the peer review history of their article (what does this mean?). If published, this will include your full peer review and any attached files.

Reviewer #1: No

Reviewer #2: No

Reviewer #3: No

---

## [Decision Letter · Decision Letter 1]

13 Apr 2020

Dear Dr Okada,

Thank you very much for submitting your Research Article entitled 'Behavioral and brain- transcriptomic synchronization between the two opponents of a fighting pair of the fish Betta splendens' to PLOS Genetics. Your manuscript was fully evaluated at the editorial level and by independent peer reviewers. The reviewers appreciated the attention to an important topic but identified some aspects of the manuscript that should be improved.

We therefore ask you to modify the manuscript according to the review recommendations before we can consider your manuscript for acceptance. Your revisions should address the specific points made by each reviewer.

[LINK]

Yours sincerely,

Alison Bell

Guest Editor

PLOS Genetics

Gregory Barsh

Editor-in-Chief

PLOS Genetics

The paper by Vu et al is much improved since its earlier version. It was re-reviewed by two of the original reviewers, and both of them were impressed by the way the authors incorporated their suggestions.

Reviewer #3 noted just a few places in the MS that could be improved. The authors should address these concerns when they revise their manuscript.

The authors should also please be sure to deposit their data in a depository.

I have just a few minor editorial suggestions, detailed below.

Line 42: please replace “higher” with “vertebrate”

Line 72: females also fight. Please remove “male” and change “conspecific” to “conspecifics”

Line 109: remove “excellent”

Line 173: change “discover” to “discovery”

Line 428: remove “genomics”; integrative genomics typically refers to integrating different types of ‘omics data, e.g. RNASeq, DNASeq, ATAC-Seq, etc

Reviewer's Responses to Questions

**Comments to the Authors:**

Reviewer #1: I have reviewed this revised version of the manuscript by Vu et al and find it much improved. They have addressed my main concerns carefully. In particular, I appreciate the addition of a clear explanation of why they hypothesized that there could be synchronization at the gene expression level in the brain (new paragraph in introduction). They have also modified several of their statistical and transcriptomic analyses and improved them, for example with permutation analyses or by having a more appropriate criteria for gene selection. Their revision goes well beyond cosmetic changes and they have made sure to redo analyses if needed, figures, etc. I think the new figure 5 is very useful and illustrates well the comparison of covariation of transcriptomes of males within a pair and between pairs.

Reviewer #3: The authors have done a very thorough job addressing the reviewers' comments! This is very well executed study that will be of interest to a broad readership. I have a few small suggestions the authors may want to consider:

1. I suggest the authors provide a couple of clear hypotheses or predictions before they launch into the Results. This would very much help with the overall framing of the study.

2. Because linear discriminant analysis (LDA) is supervised, it might be useful to also use principal component analysis (PCA) on this dataset, especially given that the authors did not have a clear expectation on the clustering.

3. The rationale for using whole brain is improved, though it seems largely based on hindsight, given the results of this study. I still think the authors made an appropriate decision here, but they can justify it better by pointing at previous studies that did whole brain gene expression profiling in the context of aggression or social status.

4. On lines 391ff. the authors now address the asymmetry in DEGs they discovered, suggesting “the possibility of a shift in metabolism that prefigures later neuronal plasticity in the brain.” This is an intriguing idea, though I wonder how generalizable it is (see, e.g.,

Cummings et al., ProcRSoc B 275:393, 2008, where fewer genes were highly expressed in the ostensibly more “salient” stimulus situation). Is it possible that the asymmetry in DEGs between baseline and fighting groups is a molecular manifestation of increased inhibitory control of gene expression on aggression (so as to avoid an over-escalation), as suggested by Wong & Hofmann, Encyclopedia of Life Sciences, John Wiley & Sons, Ltd: Chichester.

DOI: 10.1002/9780470015902.a0022554, 2010)? Also, how could any of these ideas be tested? This is obviously outside of the scope of this study, but I wanted to share these thoughts with the authors.

5. Also, consider including Figure S8 in the main paper. I think it is important in supporting your argument.

**Have all data underlying the figures and results presented in the manuscript been provided?**

Reviewer #1: Yes

Reviewer #3: No: Something that needs to be confirmed upon acceptance

PLOS authors have the option to publish the peer review history of their article (what does this mean?). If published, this will include your full peer review and any attached files.

Reviewer #1: No

Reviewer #3: No

---

## [Editor Report · Decision Letter 2]

5 May 2020

Dear Dr Okada,

We are pleased to inform you that your manuscript entitled "Behavioral and brain- transcriptomic  synchronization between the two opponents of a fighting pair of the fish Betta splendens" has been editorially accepted for publication in PLOS Genetics. Congratulations!

Yours sincerely,

Alison Bell

Guest Editor

PLOS Genetics

Gregory Barsh

Editor-in-Chief

PLOS Genetics

Comments from the reviewers (if applicable):

The authors have done a commendable job responding to the concerns raised in the last round of review and I am happy to recommend that the paper be accepted for publication in PLoS Genetics.

**Data Deposition**

http://datadryad.org/submit?journalID=pgenetics&manu=PGENETICS-D-19-01565R2

**Press Queries**

---

## [Editor Report · Acceptance letter]

28 May 2020

PGENETICS-D-19-01565R2 

Behavioral and brain- transcriptomic synchronization between the two opponents of a fighting pair of the fish Betta splendens 

Dear Dr Okada, 

We are pleased to inform you that your manuscript entitled "Behavioral and brain- transcriptomic synchronization between the two opponents of a fighting pair of the fish Betta splendens" has been formally accepted for publication in PLOS Genetics! Your manuscript is now with our production department and you will be notified of the publication date in due course.

With kind regards,

Matt Lyles

PLOS Genetics

On behalf of:
